# A Compact Rayleigh Autonomous Lidar (CORAL) for the middle atmosphere

Bernd Kaifler[1] and Natalie Kaifler[1]

[1]Deutsches Zentrum für Luft- und Raumfahrt, Institut für Physik der Atmosphäre, Oberpfaffenhofen, Germany

**Correspondence:** Bernd Kaifler (bernd.kaifler@dlr.de)

**Abstract.** The Compact Rayleigh Autonomous Lidar (CORAL) is the first fully autonomous middle atmosphere lidar system to provide density and temperature profiles from 15 km to approximately 90 km altitude. From October 2019 to October 2020 CORAL acquired temperature profiles on 243 out of the 365 nights (66 %) above Rio Grande, southern Argentina, a cadence which is 3-8 times larger as compared to conventional human operated lidars. The result is an unprecedented data set with measurements on two out of three nights on average and high temporal (20 min) and vertical (900 m) resolution. First studies using CORAL data have shown for example the evolution of a strong atmospheric gravity wave event and its impact on the stratospheric circulation. We describe the instrument and its novel software which enables automatic and unattended observations over periods of more than a year. A frequency-doubled diode-pumped pulsed Nd:YAG laser is used as light source and backscattered photons are detected using three elastic channels (532 nm wavelength) and one Raman channel (608 nm wavelength). Automatic tracking of the laser beam is realized by implementation of the conical scan (conscan) method. The CORAL software detects blue sky conditions and makes the decision to start the instrument based on local meteorological measurements, detection of stars in all-sky images, and analysis of European Center for Medium-Range Weather Forecasts Integrated Forecasting System data. After the instrument is up and running, the strength of the lidar return signal is used as additional information to assess sky conditions. Safety features in the software allow operation of the lidar even in marginal weather which is a prerequisite to achieving the very high observation cadence.

## 1 Introduction

Since several decades light detection and ranging (LiDAR; also spelled lidar) has been used to profile the atmosphere and retrieve information on aerosols, trace gases, and atmospheric density, temperature and wind (see e.g. Fujii, 2005). Following the invention of the laser, first observations of tropospheric clouds were reported in the early 1960s. Soon thereafter more powerful lasers and sensitive detectors lead to detection of stratospheric aerosols by lidar (e.g. Collis, 1965; Schuster, 1970). But it lasted until the early 1980s before the lidar technology was developed far enough to enable measurements of atmospheric density and temperature in the mesosphere (Hauchecorne and Chanin, 1980). In contrast to their tropospheric counterparts, the mesospheric lidars were rather complex experiments requiring a great deal of labour to set up and operate, with some systems filling entire buildings (von Zahn et al., 2000). Hence, these lidars were run only during campaigns or, e.g. in case of stations belonging to the Network for the Detection of Atmospheric Composition Change, during certain days per week when the

weather forecast looked favorable and trained operators were available. The intermittent operation not only limited the amount of data, but also made statistical studies that require a dense temporal sampling of the atmosphere, e.g. the investigation of the evolution of atmospheric gravity wave events (e.g. Kaifler et al., 2020), next to impossible. Gravity wave climatologies which do not require a dense sampling were published by e.g. Wilson et al. (1991); Sivakumar et al. (2006); Rauthe et al. (2008); Li et al. (2010); Mzé et al. (2014); Kaifler et al. (2015b).

In recent years a number of autonomous tropospheric lidar systems have been developed to address the shortcomings of the earlier manually operated instruments and increase the data output (Goldsmith et al., 1998; Reichardt et al., 2012; Strawbridge, 2013; Engelmann et al., 2016; Strawbridge et al., 2018). But until today, to the knowledge of the authors, no attempts were made to build autonomous middle atmosphere lidars. There may have been several factors contributing to the stalled development. First, lidars capable of sounding the mesosphere require a much higher sensitivity given the exponential decrease in air density with altitude. Consequently, mesospheric lidars use powerful lasers, large aperture receiving telescopes and highly efficient receivers, which makes some of the solutions generally used in the development of autonomous tropospheric lidars impractical, as for instance a window covering the telescope to protect it from the environment. Second, because of the technical challenges and lower interest in the middle atmosphere as compared to the troposphere, there are only a few groups operating middle atmosphere lidar instruments.

The primary objective of the Compact Rayleigh Autonomous Lidar (CORAL) is the demonstration of a fully autonomous lidar system which can be used for studying atmospheric dynamics in the stratosphere and mesosphere. That the instrument should be capable of fully automatic observations was not seen as a practical feature, but rather as pure necessity resulting from lack of human effort to operate the instrument. For the same reason, the instrument should require only a bare minimum of maintenance work. In other words, the instrument should happily sit by itself, monitor itself, and collect atmospheric measurements whenever weather conditions allow optical soundings. Human interaction should be limited to approximately weekly downloads of scientific data and yearly maintenance. Moreover, CORAL should be transportable, fully independent of infrastructure expect for electrical power, robust enough to withstand environmental conditions from the tropics to the Arctic and Antarctic, easy to replicate, and in relative terms low cost. In short, we wanted to develop a lidar system which can be set up at some remote location and left there for years collecting atmospheric data, much like ceilometers are used today by the weather services. If such a system was possible, it would surely mark the transition from the conventional, laboratory style and labor-intensive lidar systems commonly in use today and run by lidar experts, to a new generation of operational lidar systems which can be run by experts and non-experts alike. There are several benefits expected from such a new generation of lidars:

1. As the cost of lidar operators contribute significantly to the operating costs of conventional lidars, the use of autonomous systems will bring the cost per operating hour down. Lower costs will enable a more widespread use of lidar systems for atmospheric research and climate monitoring.

2. Not having to rely on human operators to acquire soundings facilitates the collection of large and continuous data sets, thus offering new possibilities for statistical analysis of the temperature structure on timescales from years to minutes.

| Laser | 120 mJ pulse energy at 532 nm wavelength, 100 Hz pulse rate |
|---|---|
| Telescope | 63.5 cm diameter, 361 µrad field of view, 60 µrad ($2\sigma$) spot size |
| Receiver | 3 elastic channels and 1 Raman channel (608 nm wavelength) |
| Data acquisition | Single pulse acquisition with 800 ps (1.2 cm) resolution |
| Data products | Temperature profiles 15-90 km altitude; resolution 900 m × 20 min (vertical × time) |
| | Higher resolutions possible for reduced altitude ranges |

**Table 1.** A summary of the lidar technical specifications.

3. A computer in charge of operating the lidar removes any sampling biases caused by the work schedule of human operators, for example less measurements during weekends and holidays.

Given these compelling advantages, it is almost incomprehensible why, in the past, little efforts have been undertaken to develop autonomous middle atmosphere lidar systems. One of the reasons is certainly that lidar scientists and engineers are often not well trained in software design and software development. As we will show later, it takes considerable efforts and time to develop and test the software required for autonomous lidar operation. In case of CORAL, the hardware of the instrument is rather unexceptional, and it is indeed the software which contains most of the complexity of the system. Another reason is that operators are required for safety reasons at some sites.

The purpose of this paper is threefold. First, we want to demonstrate the functionality of an autonomous operated middle atmosphere lidar. Second, given the advances in computer power and software development tools, this paper shall demonstrate that building of an autonomous lidar instrument is not overly complicated. And third, as we will argue in the discussion (section 5), the large and continuous data sets produced by autonomous instruments facilitates advances in science that are hardly possible with conventional human operated lidar instruments. Following this agenda, we describe the lidar instrument in section 2 followed by the description of the software used for autonomous lidar operation in section 3. In section 4 we briefly discuss our implementation of the temperature retrieval.

## 2    The lidar instrument

Development of CORAL started in 2014 as a copy of the German Aerospace Center's first mobile middle atmosphere lidar system TELMA (temperature lidar for middle atmosphere research), which was employed with much success during the Deep Propagating Gravity Wave Experiment (DEEPWAVE) field campaign in New Zealand (Fritts et al., 2016; Kaifler et al., 2015a; Ehard et al., 2017; Taylor et al., 2019; Fritts et al., 2019). CORAL measures atmospheric density in the altitude range 15-95 km and thus covers most of the stratosphere and mesosphere. The system uses a pulsed laser with 532 nm wavelength as light source and a receiver equipped with several channels for detecting both elastic scattering and inelastic scattering at 608 nm wavelength. Atmospheric temperature is retrieved by hydrostatic integration of the measured density profiles (Hauchecorne and Chanin, 1980).

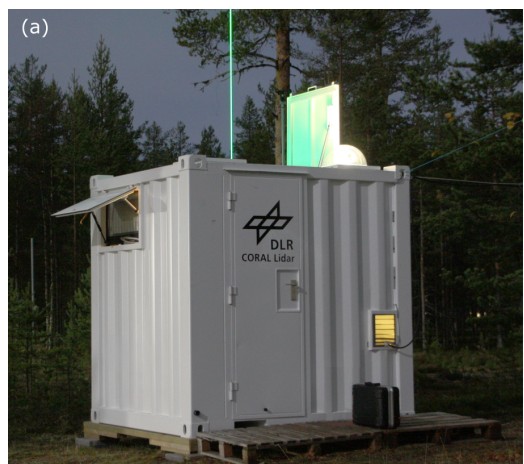 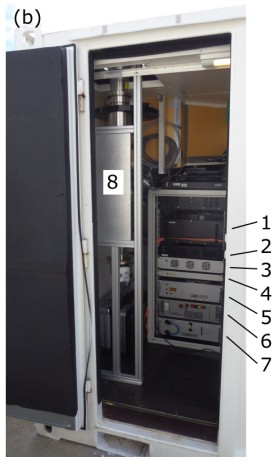 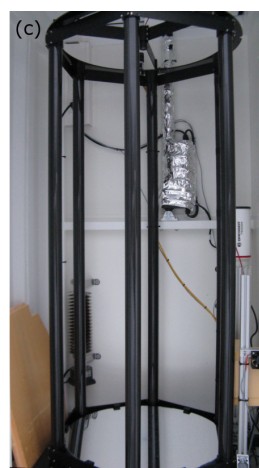

**Figure 1.** (a) A picture of the CORAL instrument container taken during lidar measurements at the Geophysical Observatory in Sodankylä, Finland in September 2015. (b) A view into the container through the open front door showing the lidar rack with the receiver (1), data acquisition computer (2), lidar electronics (3), telescope electronics (4), laser head (5), laser power supply (6), laser cooler (7), and an Advanced Temperature Mapper (AMTM) as guest instrument (8). (c) The telescope in the back of the container. Pictures by N. Kaifler.

The lidar instrument is integrated into an 8-foot steel container (see Fig. 1), which serves both as transport container and enclosure during lidar operation. The container is divided into two compartments: an air-conditioned room accommodates the transmitter, receiver and data acquisition systems, while the telescope is located in a separate room with a large hatch in the roof for direct access to the sky. The technical specifications of the lidar instrument are summarized in Table 1.

## 2.1 Transmitter

Figure 2 shows the schematics of the optical paths inside the lidar instrument. The laser (Spitlight DPSS 250-100 from Innolas GmbH) is a diode-pumped Nd:YAG master oscillator power amplifier system operating at 1064 nm wavelength and 100 Hz pulse repetition frequency. It delivers 120 mJ per pulse after conversion to the second harmonic at 532 nm wavelength. The remaining infrared light is separated and subsequently dumped using a dichroic mirror and water-cooled beam dump. A folding mirror mounted on a fast tip/tilt piezo actuator with 7 mrad angular range directs the green beam towards a 2x beam expander, which reduces the beam divergence to approximately 170 µrad and the pointing jitter to $< 50$ µrad full angle. The resulting effective beam divergence is thus $< 220$ µrad or approximately half of the telescope field of view (FOV). Finally, the beam exits the laser box through an anti-reflection coated window. A motorized mirror located in the telescope compartment of the container directs the laser beam into the sky at a position that is approximately 0.4 m offset from the optical axis of the receiving telescope.

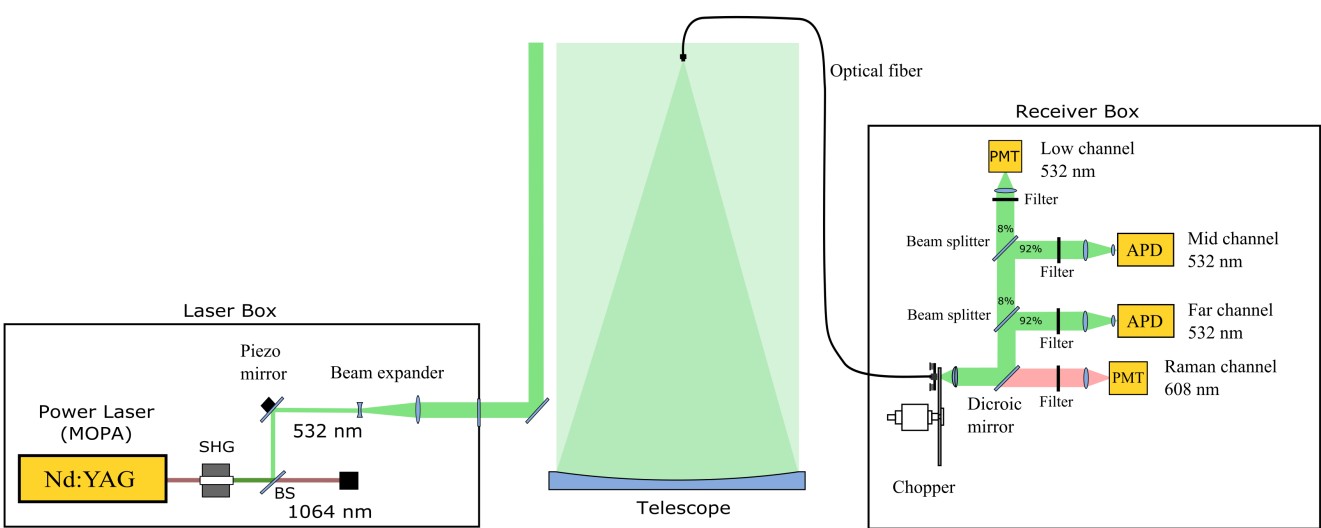

**Figure 2.** Schematics of the lidar instrument and optical paths.

## 2.2 Receiver

The backscattered light is collected using a 63.5 cm diameter parabolic f/2.4 mirror with a spot size of ~60 μm. An optical
fiber (type FG550LEC; 550 μm core diameter, 0.22 NA) mounted in the focal plane guides the collected light to the receiver.
The fiber mount consists of a spring-loaded piston traveling inside a fixed tube with the piston pushing against a linear motor
(Thorlabs Z812). With the help of the motor, the position of the fiber end can be adjusted in z-direction with ~2 μm resolution,
thus facilitating easy adjustment of the telescope focus. The outer tube is held by a three-legged spider mounted on an aluminum
ring with a diameter slightly larger than the mirror. The ring is supported by 6 vertical carbon fiber tubes that connect it to the
base plate holding the telescope mirror, and the whole telescope assembly sits on adjustment screws that allow the telescope
to be pointed to zenith. The use of carbon fiber tubes results in a high thermal stability of the telescope. A 50 K change in
temperature causes the focal point to shift by 160 μm in the vertical. As shown in the supporting information, this shift has a
negligible impact on the fiber coupling performance. During the setup of the instrument, the optimum position is determined
by slowly moving the fiber end up and down using the motor and recording the strength of the lidar return signal integrated
over the altitude range 40-50 km as function of motor position. Repeated scans are performed and measurements averaged in
order to reduce the effect of potential changes in atmospheric transmission during the scans. The focal position is determined
as the position with the maximum lidar signal.

With the bistatic setup full geometric overlap between the laser beam and the telescope FOV is achieved at approximately
5 km altitude. However, range-dependent defocusing of the telescope causes the overlap function to vary slightly with altitude.
This variable overlap results in a bias in retrieved temperature profiles of less than -0.4 K for altitudes above 40 km and
increases to -0.95 K at 15 km altitude (see section S2 in the supporting information).

The optical bench of the receiver resides in a four-units 19-inch rack mount enclose. The optical fiber enters the enclosure at the back side and terminates in front of a mechanical chopper with three slits rotating at 100 revolutions per second. The firing time of the laser is synchronized with the rotation of the chopper such that laser light scattered in the lower 14 km of the atmosphere is blocked by the chopper blades and does not hit the sensitive detectors. As shown in Fig. 2, after passing through the collimation optics, the collimated beam is spectrally divided into two parts by a dichroic mirror, separating the elastic scattering at 532 nm wavelength and the nitrogen rotational Raman scattering at ∼608 nm wavelength. The Raman scattering is detected using a photomultiplier (Hamamatsu H7421; approximately 35 % detection efficiency at 600 nm) with a 3 nm wide interference filter (80 % peak transmission, out-of-band blocking optical depth (OD) >6) mounted in front. The dichroic mirror has a transmission of 1.2 % at 532 nm wavelength and this results in a total blocking of the elastic scattering in the Raman channel of OD ∼8.

In order to increase the dynamic range of the detection, the beam containing the elastic scattering is further split into three beams with a splitting ratio of approximately 92.0:7.4:0.6, i.e. the detector of the far channel sees 92 % of the light, while only 0.6 % of the light reaches the low channel. Both the high and mid channel detectors are avalanche photo diodes (APDs) operated in Geiger mode (SPCM-AQRH-16 from Excelitas; ∼50 % detection efficiency at 532 nm wavelength) with 0.8 nm wide interference filters (83 % peak transmission) mounted in front. The APDs are gated to limit peak count rates to about 5 MHz. The low channel detector is again a photo multiplier tube (Hamamatsu H12386-210) with a 3 nm wide cost-efficient interference filter (60 % peak transmission).

## 2.3   Data acquisition and control

The data acquisition system comprises three units, the acquisition computer, the control electronics, and the MCS6A photon counter. The MCS6A produced by FAST Comtec GmbH is a five-channel multi-event digitizer with 800 ps resolution. It converts the electrical pulses coming from the detectors to timestamps indicating the elapsed time since the firing of the laser. The data acquisition software running on the computer reads out the MCS6A after each laser pulse and stores the timestamps on solid state drives for post-processing as well as sorts them into histograms for displaying the photon count profiles in real-time.

Trigger signals for the laser, chopper and APD gating are produced by the control electronics. Its core is a National Instruments SBRIO-9633 embedded single-board computer with a field-programmable gate array (FPGA). The trigger chain of the lidar is implemented in the FPGA and programmable delays in the outputs allow for adjusting the timing between the signals, for example setting the phase and thus the opening altitude of the chopper and controlling the gating of the APDs. Analog outputs of the SBRIO drive the tip/tilt piezo mirror inside the laser through high voltage amplifiers also located in the electronics box. Prior to outputting the analog signals, the drive signals for the piezo mirror are conditioned and limited in bandwidth by digital filters implemented in the FPGA to prevent the mirror from overshooting the target position and excitation of resonant modes. Finally, the electronics box also houses the power supplies for the detectors and relays that are controlled by the FPGA for switching the detectors on and off.

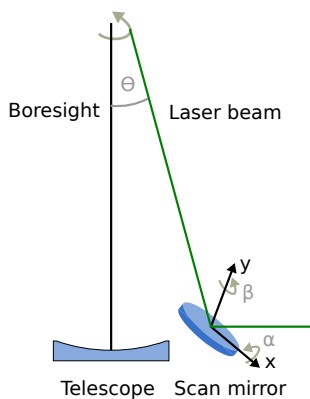

**Figure 3.** The coordinate system of the scan mirror.

### 2.3.1 Automatic tracking of the laser beam

One problem with container-based lidar systems is the limited thermal stability. When the telescope hatch opens and the telescope compartment cools down, thermal drifts result in misalignment between the telescope boresight and the laser beam. This drift is especially problematic for lidar systems which use narrow field of views in the order of few hundred microradian for low background noise, and active tracking of the laser beam position is usually required. Two methods are commonly used to track the laser beam. With manual tracking, an operator performs a quadrant search by moving the laser beam by a small

angle in alternating directions while monitoring the strength of the lidar return signal. When the scan is completed, the beam angles which yield the strongest lidar signal are chosen as the new beam position. The search is repeated at regular intervals e.g. hourly. Innis et al. (2007) describe an automatic autoguiding system which uses a camera looking through the receiving telescope to image the spot of the laser beam in the atmosphere at a certain altitude. A software analyzes the images and computes the beam position. Any deviation from the predetermined target position is nulled by a servo loop. The target needs

to be found by other means e.g. manual search. While the latter method is suitable for an automatic lidar like CORAL, we opted to implement the conscan method that is widely used for tracking spacecraft (see e.g. Gawronski and Craparo, 2002). It's main advantages over the autoguiding method (Innis et al., 2007) are the possibility to evaluate the lidar signal for tracking purposes in the stratosphere and thus well above any potential cloud layer, and no requirement for a predetermined target position. In particular the resilience to clouds is attractive, as CORAL is designed to operate in marginal weather conditions.

To our knowledge, this is the first application of conscan to mesospheric lidars.

The basic principle of conscan is depicted in Fig. 3. A scan mirror rotates about the axes x and y in a sinusoidal motion, causing the laser beam to rotate around the telescope boresight in a conical scan (see Fig. 4). If the center of the cone is offset from the boresight of the telescope, the angle between the laser beam and the telescope boresight $\Theta$ periodically becomes smaller and larger due to the conical motion of the laser beam. Assuming the offset of the cone is sufficiently large, the

modulation of $\Theta$ leads to an incomplete overlap between the telescope FOV and the laser beam. This in turn causes a modulation in the signal strength of the lidar return signal, which can be demodulated and the information used to infer the direction the

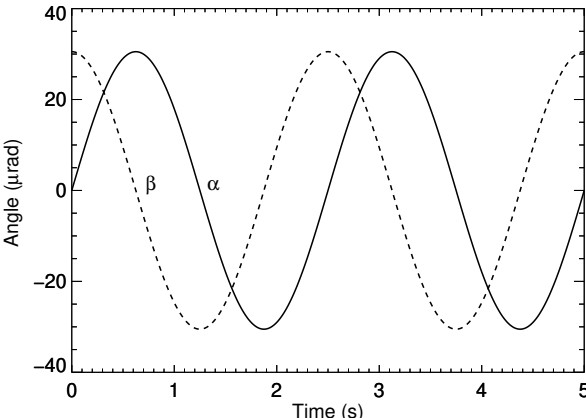

**Figure 4.** Angular modulation of the scan mirror.

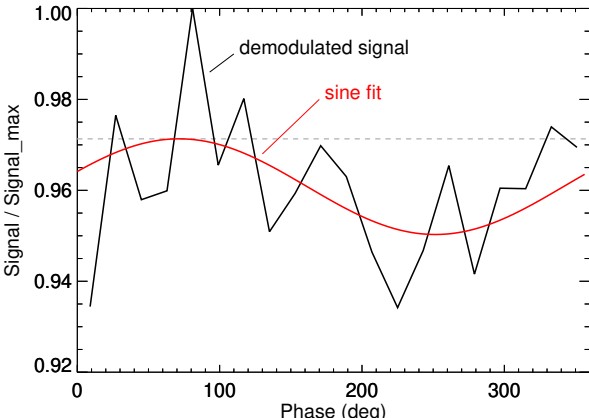

**Figure 5.** Example of a demodulated conscan signal acquired with CORAL. The dashed line indicates the mean signal that would be obtained in case of a perfect beam overlap and no modulation.

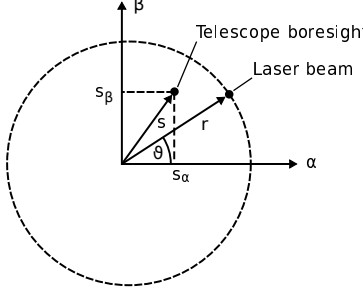

**Figure 6.** Position of the laser beam and telescope boresight during a conscan (adapted from Gawronski and Craparo, 2002).

axis of the cone needs to be shifted to in order to obtain a complete overlap. An example of such a demodulated signal is shown in Fig. 5. Looking at the geometry depicted in Fig. 6, it becomes clear that maximum overlap is achieved if vector $\boldsymbol{r}$ points in the same direction as vector $\boldsymbol{s}$. The corresponding direction in the coordinate system of the scan mirror is given by the vector

$$\boldsymbol{e}_{\mathrm{s}} = \begin{pmatrix} \cos\vartheta_{\mathrm{s}} \\ \sin\vartheta_{\mathrm{s}} \end{pmatrix} \tag{1}$$

where $\vartheta_{\mathrm{s}}$ denotes the phase angle with the largest demodulated lidar return signal.

Equation (1) tells us in which direction we have to move the laser beam in order to achieve complete overlap, but we don't know how far along $\boldsymbol{e}_{\mathrm{s}}$ we have to go in order to reach the point defined by $\boldsymbol{s}$. Based on the data at hand, there is no way to determine the scaling factor $l$ in the relation $\boldsymbol{s} = l\boldsymbol{e}_{\mathrm{s}}$, but we can estimate $l$ from the amplitude of the conscan signal. For
simplicity, we initially assume a perfect lidar producing noise-free measurements. Let's consider the situation where the mean $\Theta$ equals half of the telescope FOV and the amplitude of the modulation signal driving the conscan, $|\boldsymbol{r}|$, is so large that the lidar return signal oscillates between zero (no overlap) and a maximum (complete overlap) and the demodulated conscan signal, in the following denoted as $A$, shows oscillations between zero and one. This can be achieved only if $|\boldsymbol{r}|$ also equals half of the telescope FOV. When the modulation amplitude or mean $\Theta$ are smaller, the minimum lidar return signal must be larger than
zero as there is always a partial overlap. In this case $A$ contains a nonzero offset $c$ and the amplitude of the sinusoidal part $a$ is smaller than one, and we can rewrite $A$ as

$$A = c + a\cos\left(\vartheta - \vartheta_{\mathrm{s}}\right) \tag{2}$$

On the other hand, for the other extreme case where the conscan modulation and mean $\Theta$ are so small that the laser beam is always completely inside the telescope FOV, we expect no variation in $A$ and hence $\alpha = 0$. Thus, the amplitude $a$ can be used
as an estimate of the overlap. For simplicity, in the following we assume a linear relationship and approximate $\boldsymbol{s}$ as

$$\boldsymbol{s} \approx \hat{\boldsymbol{s}} = \begin{cases} 10\,a\,|\boldsymbol{r}|\,\boldsymbol{e}_{\mathrm{s}} & \text{if } a < 0.1 \\ |\boldsymbol{r}|\,\boldsymbol{e}_{\mathrm{s}} & \text{otherwise} \end{cases} \tag{3}$$

The factor 10 in the first case facilitates faster convergence when the overlap is almost complete ($a$ is small). We note that a more accurate relation can be derived from calculation of the geometric overlap function based on the actual beam profile of the laser, but the approximation in Eq. (3) is sufficient for our purposes. After a conscan is completed, the orientation of the
conical scan is updated by adding $\hat{\boldsymbol{s}}$ to the current orientation and a new conscan is started. This cycle of scanning and updating of the beam pointing is constantly repeated during the lidar measurement, causing the mean position of the laser beam to track the telescope FOV.

In our implementation of conscan we set $|\boldsymbol{r}| = 43$ µrad and the speed of the conical motion to 0.4 Hz. Furthermore, we bin the conscan signal $A$ using bins of $18°$ width and average it over 20 s i.e. 8 revolutions of the laser beam and 100 laser
pulses per bin. The averaging reduces the impact of statistical variations in atmospheric transmission (e.g. caused by clouds) and fluctuations in laser power. Figure 5 shows such an averaged conscan signal that was acquired on 3 November 2019

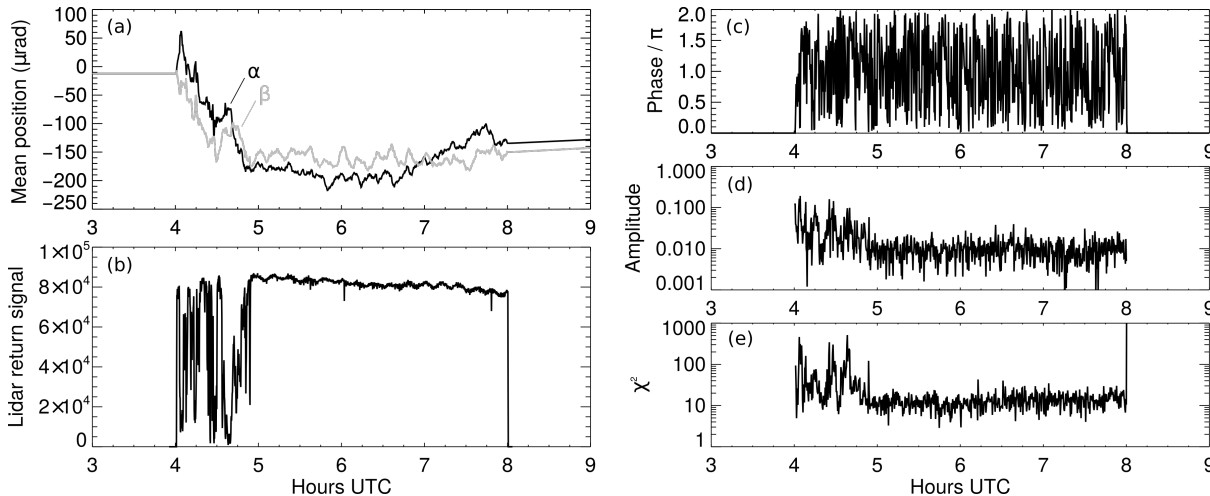

**Figure 7.** Performance of the conscan system during the measurement on 3 November 2019. (a) Mean scan mirror angles, (b) lidar return signal integrated between 45 km and 55 km altitude, (c) phase and (d) amplitude of the demodulated conscan signal, and (e) goodness of the fit.

05:15 UTC in the altitude range 45-55 km using the far channel detector. The demodulated signal contains a significant noise portion, but a sinusoidal modulation with a maximum at about 75° is nevertheless evident. In order to get a better estimate of the amplitude and phase of the maximum, we perform a sinusoidal fit using the MPFIT algorithm (Markwardt, 2009). For the example shown in Fig. 5 we obtained values $a = 0.0105$ and $\vartheta_s = 71.5°$, which according to Eq. (3) cause a shift of 3.5 µrad towards the telescope boresight when the conscan algorithm is executed. Figure 7 shows mean angles of the scan mirror for a 4-hour long lidar measurement. After startup of the instrument, warming-up of the laser and cooling-down of the telescope compartment of the container caused a drift of about 300 µrad (distance in both axes) during the first hour. That is significant compared to the telescope FOV of 361 µrad and would lead to dramatic losses in the lidar return signal if no beam tracking were used. However, as shown in Fig. 7, with beam tracking enabled the lidar return signal remained fairly stable throughout the lidar measurement. Note that while the lidar return signal was impacted by broken clouds during the first hour, yet conscan allowed robust beam tracking as indicated by the peaks in the lidar return signal reaching values of $\sim 8 \times 10^4$ which is approximately the same value as later when the clouds disappeared.

Panels c-e of Fig. 7 show the phase, amplitude and $\chi^2$-value determined from the fits to the conscan signal. The $\chi^2$-value is used as an indicator for the quality of the conscan signal. Note that we do not normalize the conscan signal prior to fitting the data and hence, the average $\chi^2$-value is about 20 instead of unity even for signals with a large signal-to-noise ratio. A large $\chi^2$ value, we use 300 as threshold, indicates that the conscan signal could not be properly demodulated. In this case the conscan cycle is aborted and the beam pointing not updated. In cloudy conditions as many as 9 out of 10 conscans may fail in that way, but the succeeding conscans are still sufficient for beam tracking, as thermal drifts happen on relatively long time scales. If more than 10 successive conscans fail, subsequent intervals are marked in the raw data files of the lidar as potentially having

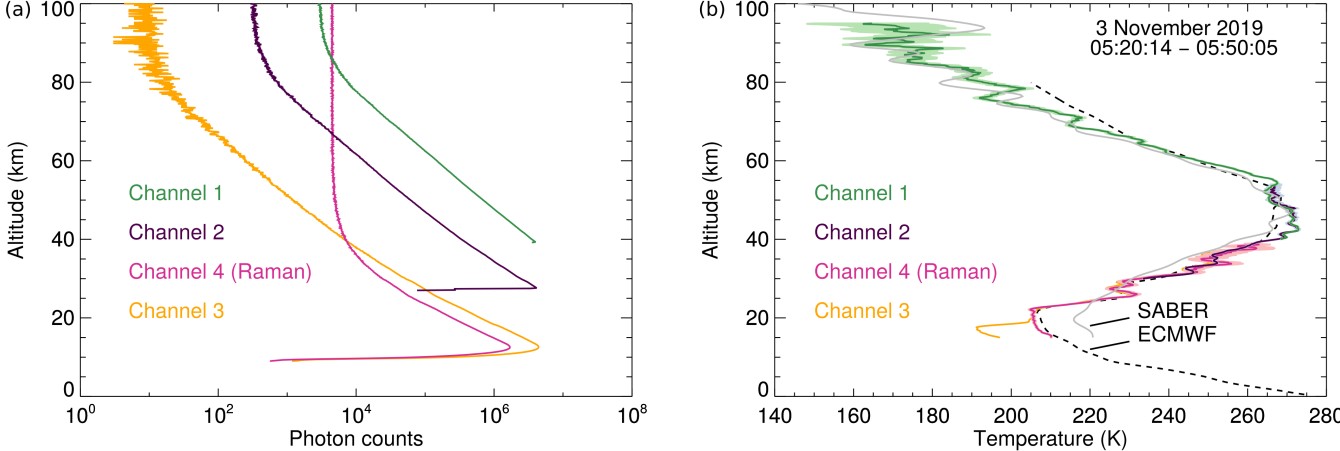

**Figure 8.** (a) Photon count profiles acquired on 3 November 2019 between 05:20 and 05:50 UTC and binned to 100 m vertical resolution, and (b) retrieved temperature profiles with 900 m vertical resolution. The shaded areas indicate the temperature uncertainties as determined by the retrieval (see section 4). For comparison, the corresponding SABER profile (acquired at 525 km mean distance to the west of CORAL at 05:38 UTC) and 06:00 UTC ECMWF IFS profile (0.125° resolution, grid point 53.75°S 67.75°W, 3.9 km distance to CORAL) are also shown as gray and black dashed lines, respectively.

incomplete beam overlap and are discarded in the temperature retrieval. Moreover, beam pointing increments and conscan parameters are stored in raw data files for offline analysis. As evident from Fig. 7e only few conscan measurements exceed the threshold of $\chi^2 > 300$ even though the lidar return signal was strongly impacted by clouds in the period 4-5 UTC. The larger uncertainties in that period are also reflected in the larger amplitude estimates (Fig. 7d).

It is important to note that conscan always evaluates the lidar signal within a constant altitude range in the stratosphere. This is in particular beneficial for bistatic lidars where in an autoguiding setup (Innis et al., 2007) the camera images the laser beam at an angle in the lower troposphere, as scattering in the stratosphere is too weak for imaging. Due to the finite exposure time, the camera integrates the beam profile along a certain height range. If a cloud drifts through the beam within this height range, the image will be dominated by the strong Mie scattering. If that happens at the bottom (top) of the height range, camera based

autoguiding optimizes the beam overlap for low (high) altitudes in bistatic lidar configurations. The use of conscan avoids that problem.

### 2.3.2  Adaptive detector gating

As one of the goals of CORAL is to obtain measurements as often as possible, it was clear from the very beginning that CORAL would also operate in marginal weather e.g. haze and variable cloud cover. Although these conditions diminish the

lidar return signal, being a Rayleigh lidar, CORAL has still enough power margin to produce scientifically usable measurements in the stratosphere and lower mesosphere. However, the weaker signal requires that gating of the APDs and the opening of the

chopper have to be adjusted to lower altitudes to make use of the full dynamic range of the detectors and allow for assembly of the individually retrieved temperature profiles into a single continuous profile (see Fig. 8).

Our implementation of adaptive controls for detector gating is rather simple. The data acquisition software integrates photon counts for two-second intervals and calculates peak count rates for each detector channel. If the peak count rate is outside a predefined dead band, the delay of the gating signal for the respective channel is increased by 3 µs if the count rate is high, or decreased by 3 µs if the count rate is low. The change is equivalent to an increase or decrease of the gating altitude in steps of 450 m. We use different dead bands $[4.0\,\mathrm{MHz}, 5.5\,\mathrm{MHz}]$, $[5.5\,\mathrm{MHz}, 6.5\,\mathrm{MHz}]$, $[8.0\,\mathrm{MHz}, 9.0\,\mathrm{MHz}]$ for the far channel, mid channel, and low channel, respectively. Lowest peak count rates are reserved for the far channel in order to limit thermal heating of the APD and thus reduce nonlinear effects that may strongly affect retrieved temperatures at upper mesospheric altitudes where the lidar return signal is low. Nonlinear effects at low count rates are of less importance in case of the other channels because, at the top of the profiles, there is sufficient overlap with temperature profiles retrieved from other channels.

### 2.3.3 Air traffic safety

Eye safety is an important consideration for an automatic lidar system. The laser beam emitted by CORAL can be dangerous for eyesight even at altitudes of aircrafts if one looks into the beam, and measures must be taken to avoid accidental exposure. A common method is to employ a radar to track aircrafts and automatically block the laser beam or shut down the laser when an aircraft approaches the lidar station (see e.g. Strawbridge, 2013). At the southern tip of South America air traffic is so low that no particular safety measures are requested by the authorities in Argentina. We use an Automatic Dependent Surveillance-Broadcast (ADS-B) receiver for receiving position data of nearby aircraft. This information is used by the lidar computers to automatically shut down the laser when the aircraft enters a circle of 800 m radius centered at the location of CORAL, and measurements are resumed when the aircraft has exited the circle. However, due to the low air traffic and CORAL being located close to the local airport but outside the flight corridors, within the three years of operation no single aircraft was detected within the critical zone. At other sites in Europe, we employed a combination of an ADS-B receiver and a camera-based setup for tracking position lights of aircrafts. Because CORAL operates only in darkness, the bright position lights result in a good signal-to-noise ratio in the camera images and can be easily tracked using motion detection algorithms. For protection against software crashes, a kill timer, reset by the ADS-B decoder and optical tracker software, is implemented in the FPGA controlling the laser. In case the software crashes, the kill timer runs out and the laser is automatically shut down after 3 s.

### 2.4 Container

The CORAL container provides all the necessary infrastructure for the operation of the CORAL lidar instrument. It has two large doors, one in the front providing access to the air-conditioned compartment and one in the back for servicing the telescope (see Fig. 9). Two smaller hatches equipped with actuators serve as inlet and outlet for the air needed by the chiller. A third motorized hatch of size 0.8 m by 0.8 m is located above the telescope (see also Fig. 1). Finally, two smaller openings of size 0.4 m by 0.4 m in the roof enable the installation of transparent domes for passive optical instruments. While one dome is usually occupied by a cloud monitoring all-sky camera, the other dome is available to guest instruments such as the

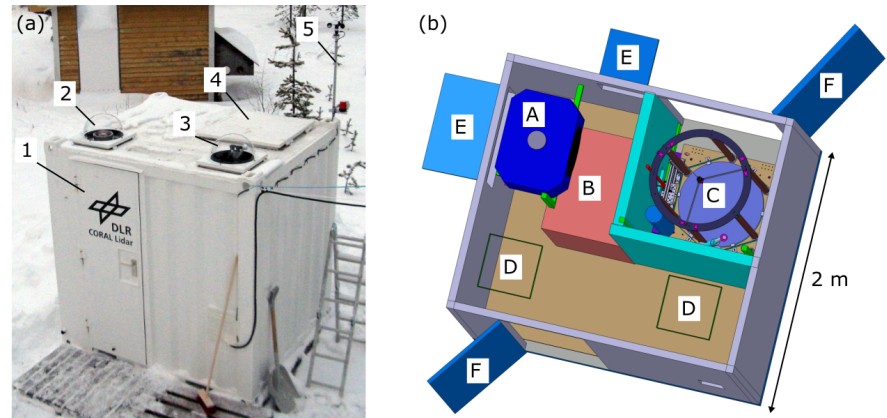

**Figure 9.** (a) The CORAL container at the Geophysical Observatory in Sodankylä, Finland with access door (1), optical dome for passive instruments (2), optical dome for all-sky cloud camera (3), telescope hatch (4), and weather station (5). (b) Layout of the interior with the water chiller (A) (no air ducts shown), lidar rack (B), telescope (C), space for passive optical instruments (D), motorized chiller hatches (E) and access doors (F). Picture by B. Kaifler.

Advanced Mesospheric Temperature Mapper (Pautet et al., 2014; Reichert et al., 2019). The domes can be removed and covers installed to seal the openings for shipment of the container. A weather station measuring wind speed, temperature, humidity and precipitation completes the external additions.

    The layout of the interior is sketched in Fig. 9b. The larger of the two compartments is insulated and air-conditioned to $22\pm2°$C, whereas the smaller telescope compartment is only equipped with low-power electrical heaters to raise its temperature 275   slightly above the ambient temperature in order to reduce the humidity when the lidar is not in operation and the telescope hatch closed. The chiller, which is mounted below the ceiling, provides cold glycol with a cooling capacity of 2.4 kW and is used for booth secondary cooling of the laser and air conditioning. All of the lidar hardware with exception of the telescope is installed in the laser rack below the chiller. The space between the two boxes marked with "D" in Fig. 9b is normally kept empty and can be used by a person for servicing the lidar or manual on-site control of the lidar.

**2.4.1   Control**

Control of the container systems such as heatings, hatch actuators, fans and chiller, is exercised by two ATMEGA644 8-bit microcontroller units on custom designed electronics boards. The microcontroller units serve as multiplexers and demultiplexers (MDMs) by providing discrete signals to relays supplying power to the various subsystems and reading data from the weather station as well as other internal sensors e.g. level sensors and temperature sensors in the glycol tank of the chiller. Aggregated 285   sensor and status data are sent to, as well as commands received from, a high-level container control computer (COCON) via serial RS-232 links. Whereas a single-task program with a global event loop runs on the bare hardware of each MDM, COCON is a standard x86-compatible computer running the Linux operating system and application software written in C/C++. The MDMs and COCON run normally in tandem. For example, COCON would send a high-level command to switch the chiller

on to MDM #1. The MDM decodes the command and translates it into a set of low-level instructions for immediate or deferred execution: commanding the actuators to open the hatches for air cooling, wait until the hatches are fully open, close the relay to supply power to the chiller. At the same time, a steady stream of sensor data is flowing from the MDM back to COCON, containing e.g. angle measurements of the hatches and the coolant temperatures.

In addition to the multiplexing and demultiplexing functionality, the software running on the MDMs also includes a basic set of fault protection routines (FPRs). The sole purpose of these FPRs is to guarantee that the CORAL system is always in a consistent and safe state. There are FPRs dealing with technical faults as well as, in the view of CORAL, dangerous environmental conditions. For example, an FPR prevents opening of the telescope hatch if the wind speed as measured by the weather station exceeds a certain threshold, and another FPR is responsible for shutting down the laser and closing of the telescope hatch when precipitation is detected. The implementation of the most critical FPRs at the MDM-level represents a safeguard against adverse effects of software errors. Because the software running on the MDMs is less than 3000 lines in total and does not rely on an intercalated operating system, the probability of a software error causing a fatal crash or deadlock is much lower than it is the case for the application software running on COCON with its hundreds of thousands of lines. In that we cannot guarantee that the high-level application software is free of errors, we *have to* assume that it fails at some point, and hence, with no operator in the loop to intervene, the MDMs must be capable of their own to maintain the safety and a consistent state of the CORAL system to prevent fatal outcomes such as leaving the hatch open in a rain shower. Following this requirement, most FPRs trigger a routine called "safe-mode" which shuts down the lidar, disables the chiller, closes hatches and reconfigures the heating and ventilation system. It is then up to the application software running on COCON to recover from the fault that caused safe-mode. Following the example with high wind speed, the application software monitors data coming from the weather station and, after the wind has sufficiently abated, restarts the lidar operation.

Another more severe example is power failure. All critical computers, sensor busses and actuators are powered off an uninterruptible power supply (UPS). The only exception is MDM #2 which, for reasons of redundancy, is directly connected to the main power supply. In the event of a power failure, MDM #2 thus shuts down. Since both MDMs are constantly monitoring each other by sending heart beat signals, MDM #1 detects power failures as absence of the MDM #2 heart beat signal and triggers the corresponding FPR. On a higher level, the application software running on COCON also monitors the state of the UPS and has access to additional information such as the battery charging level. While the FPS on MDM #1 triggers closing of all hatches in the event of a power failure, the application software may shut down the computers if the charging level becomes too low. The computers boot up automatically after power is restored.

## 3   Autonomous lidar operation

### 3.1   Software architecture

The three key ingredients that make autonomous lidar operation possible are (i) the ability to control every aspect of the lidar instrument and container subsystems by means of a computer program, (ii) the availability of robust data based on which the decision can be made whether lidar operation is currently feasible, and (iii) the implementation of this decision-making logic.

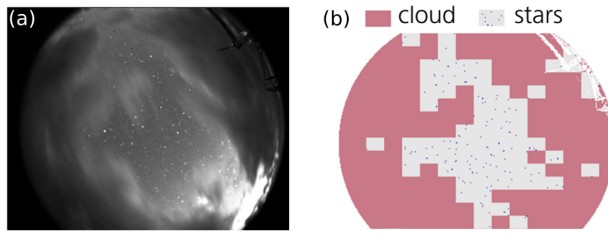

**Figure 10.** (a) Image acquired by the cloud monitoring camera and (b) detected stars. Image by N. Kaifler.

The first is a pure technical aspect which we realized by implementing a message-based data exchange system on top of a client-server architecture. The functionality of each subsystem such as lidar data acquisition, laser, and autocontrol—this part contains the decision-making logic—is implemented in separate computer programs that communicate via the message system. For example, autocontrol inquires the data acquisition about the strength of the lidar return signal, and the data acquisition reports the numbers back to autocontrol by replying to that message. In another example the autotrack program, which tracks the laser beam, requests photon count data from the data acquisition, processes the data, and sends a message to the lidar electronics to update the beam pointing. Short messages that may contain only few parameters or data values are implemented using the Standard Commands for Programmable Instruments (SCPI) protocol (SCPI), while larger data sets are sent as binary blobs preceded by a unique identifier. SCPI is a human readable protocol. For example, the command *laser:shutter 1* prompts the laser to open its shutter. All aspects of the lidar system can be controlled without the need for a graphical user interface by typing SCPI commands in a terminal. This simplified testing and debugging of the software a lot given that in total >1000 commands are currently implemented in the lidar software, the majority representing configuration parameters. SCPI commands also can be collected in text files that are loaded and sent automatically at startup of a program.

## 3.2 Data sources and decision making

The decision-making process behind the autonomous capability of CORAL is implemented in a program called autocontrol. Autocontrol is a rule-based system that seeks answers to questions such as: is it cloudy? Is the cloud layer solid (no lidar observations possible) or broken clouds (lidar observation possible, but signal degraded)? What is the probability for rain within the next hour? The individual answers are then combined in logical connections to arrive at the yes/no decision to start or stop the lidar.

In order for autocontrol to find answers, we have to feed it with data. The current implementation uses five main data sources: the solar elevation angle, the local weather station, the cloud monitoring camera, ECMWF IFS data, and lidar data if the lidar instrument is already running. The solar elevation angle is determined by the fixed location of the instrument and the local time (Montenbruck and Pfleger, 2013). Because CORAL can operate in darkness only, we use the elevation angle to restrict operation times to periods when the solar elevation is below -7°. The weather station is used for monitoring precipitation and wind speed. A rain signal or the wind speed exceeding a threshold prevents the automatic start of the lidar or, in case the lidar is already running, triggers an immediate shutdown. The cloud monitoring camera is a 1.3 mega pixel monochrome CCD camera

(Basler acA1300-30gm) combined with a 1.4 mm f/1.4 fishy eye lens (Fujinon FE185C046HA-1). We use a blob finding algorithm to detect stars in long exposures, and stars are counted within a region extending from zenith to 50° off-zenith. The star count is used by autocontrol to assess whether the sky is clear (large number of detected stars) or cloudy (low number of detected stars). An example image along with a map containing positions of detected stars is shown in Fig. 10.

Relying solely on star images to discern clear sky has the disadvantage that this information is only available when the sky is sufficiently dark for stars to be seen (solar elevation angle <-11°). However, in order to facilitate early starts of the lidar and thus maximize the run time, we need information on sky condition already at twilight. This information is retrieved from ECMWF forecast data in the form of the parameters total cloud cover and total precipitation for the grid point nearest to the location of CORAL. Lidar start is allowed if the cloud fraction is below 0.5 and accumulated precipitation within the next 2 h is below 0.1 mm.

After the lidar is up and running, the strength of the lidar return signal is used as additional information for the assessment of clouds. If the signal strength is greater than 70 % of the expected maximum signal, the sky is classified as clear and lidar operation is allowed to continue even in case of ECMWF forecasting precipitation. The reasoning behind this rule is that the predicted occurrence of rain showers is often off by more than one hour and the effect of rain showers can be very localized in the surroundings and not necessarily at the precise location of CORAL. By allowing the lidar to continue observations when the signal is good prevents unnecessary shutdowns. The idea here is that precipitation is preceded by a cloud layer that can be indirectly detected by the lidar as drop in the lidar return signal strength. Following that, the lidar is stopped in case the signal drops below 70 % and ECMWF forecasts >0.1 mm of precipitation. But even if no precipitation is forecasted, the clouds may thicken enough that continuing the lidar operation is not worthwhile anymore because of the low lidar signal. For this case we implemented a 15-min count down timer that is started when the lidar signal drops below 20 %, and the lidar is stopped when the counter reaches zero. In order to let the lidar continue its observations in broken clouds, the counter is reset to 15 min every time the signal increases beyond the 20 % threshold. A final rule introduces a mandatory a wait period of 30 min following a shutdown triggered by low signal. This rule was implemented after we discovered that light scattered off dust particles on the optical dome covering the all-sky camera is sometimes misinterpreted as stars. In some cases, the high number of artificial stars leads to a constant startup-shutdown cycle even though the sky was overcast and no meaningful lidar observations could be obtained. The implementation of the wait period reduces the number of start-attempts to a level that does not cause excessive wear.

The current state of the lidar is tracked in a global state machine where violations of the rules described above trigger state changes. Rules are evaluated once per second, though data tables may be updated at different intervals depending on the data source and how often new data become available.

### 3.3 Example

Figure 11 shows the data used by autocontrol to make start/stop decisions on the night of 23-24 August 2020. The ECMWF cloud fraction (Fig. 11b) is below 0.4 during the whole night, indicating to autocontrol that no significant cloudy periods are to be expected. Start-up of the lidar is blocked until the solar elevation angle (Fig. 11a) decreases below -7°, which happens

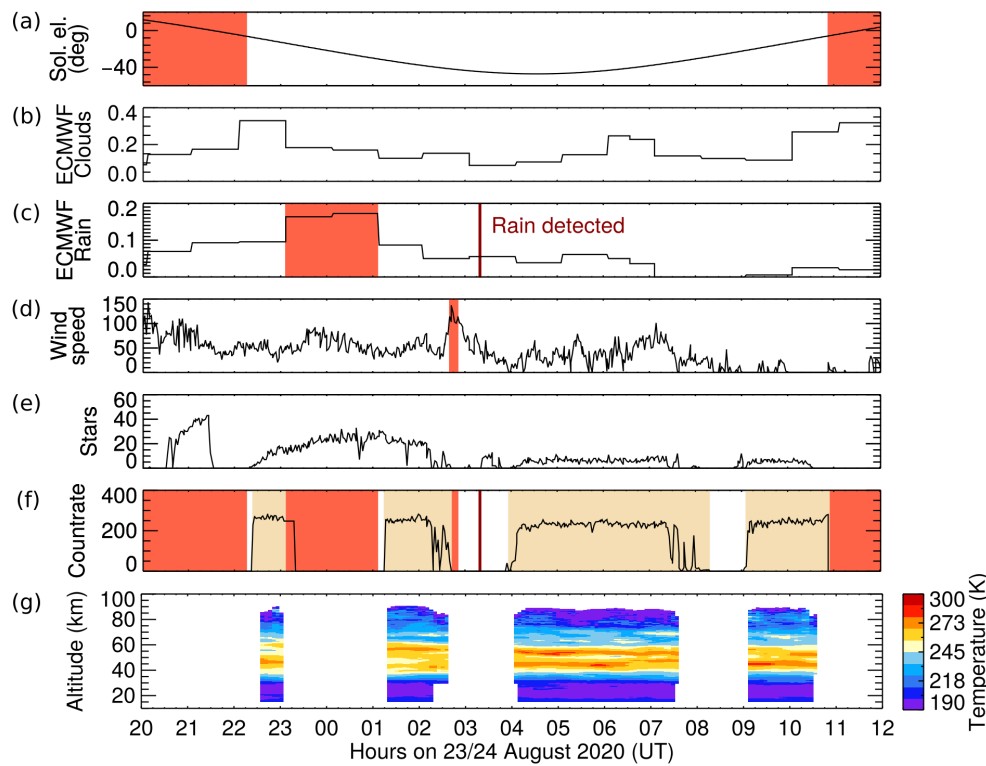

**Figure 11.** (a-f) Example data used by autocontrol to make start/stop decisions and (g) retrieved temperature profiles. Red areas mark periods with violated conditions and beige areas indicate actual instrument run times.

.

at 22:15 UTC. Then autocontrol verifies that the precipitation forecasted by ECMWF for the next two hours is below 0.1 mm (Fig. 11c), the measured wind speed is below the threshold of 110 (equivalent to approximately $15 \, \mathrm{m \, s^{-1}}$), and the rain sensor does not detect any rain. No conditions are violated and autocontrol initiates the starting sequence of the instrument. Data collection begins approximately 5 min later with photon count rates averaged between 50 km and 60 km altitude reaching about 340 kHz (Fig. 11f). At 23:08 UTC a fault protection routine within autocontrol detects the crash of the experimental data acquisition software we were testing at that time and triggered a shutdown of the instrument. The crash is evident in the photon count rate being constant, indicating that a key metric of the data acquisition software is not updated any more. At the same time, the ECMWF precipitation forecast exceeds the threshold of 0.1 mm and thus prevents autocontrol from restarting the instrument until two hours later, though the sky remains mostly cloudless as indicated by the high number of detected stars (Fig. 11e). Finally, at 02:15 UT, a simultaneous decrease in photon count rate and number of detected stars suggest the appearance of clouds. Lidar measurements continue until about 30 min later when the wind speed crosses briefly the threshold and autocontrol triggers another shutdown of the instrument. The number of detected stars remains zero while a short rain event is detected at 03:20 UTC. Although the star count increases shortly after, the start-up of the lidar is blocked by the 30 min wait

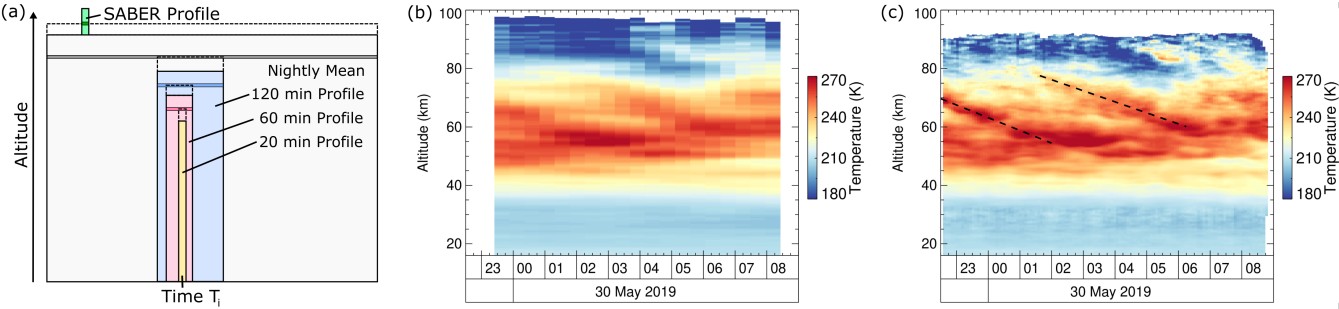

**Figure 12.** (a) Sketch of the pyramid of integration times that is built up by the retrieval when moving from longer to shorter integration periods. The horizontal bars with dark colors mark the altitude where the retrieval is seeded using the temperature profile from the preceding level. (b,c) Temperature profiles retrieved with (b) 120 min integration time and (c) 20 min integration time. Dashed lines highlight phase lines of gravity waves (see text for details).

period following a shutdown. This is a safety mechanism as it is not clear whether the nonzero star count is due to real stars being detected (cloudless sky) or due to light scattered off rain droplets on the camera dome. Half an hour later, the star count goes to zero, indicating clouds. When the star count increases again at about 04:00 UTC, autocontrol initiates the start-up of the instrument. Data collection continues until four hours later when the photon count rate and the star count reach low values. At 09:00 UTC the sky clears off again and autocontrol restarts the lidar, which then runs until the solar elevation angle increases beyond -7°.

The example shown in Fig. 11 is representative of an observation that would have kept a human operator busy throughout the night. Instead, the CORAL instrument took all decisions on its own and even recovered from a software crash without human intervention. About 2 h of data were lost due to the crash, as high photon count rates normally take precedence over the ECMWF precipitation forecast and, in this case, would have allowed the observation to continue.

## 4   Temperature retrieval

Our implementation of the temperature retrieval is based on the integration method developed by Hauchecorne and Chanin (1980). In the absence of aerosols and absorption by trace gases such as ozone, and after taking into account the geometric factor $z^{-2}$ stemming from the formation of spherical waves of the scattered light, the lidar return signal can be assumed as proportional to the Rayleigh backscatter of the atmosphere (e.g. Leblanc et al., 1998). Because Rayleigh scattering is directly proportional to the number density of air, the Rayleigh backscatter profile equates to a profile of relative number density. In the retrieval the atmosphere is split up in discrete layers and the relative number density is evaluated at these layers from layer-averages of the Rayleigh backscatter profile assuming that the number density varies exponentially across layers. Using the ideal gas law and assuming hydrostatic equilibrium, temperatures are calculated by integrating relative number densities from top to bottom. One complication arises from the requirement to start the integration at infinity where the density is zero. Since this is impractical because of the large noise in lidar measurements at very high altitudes, instead, the integration is split into

two parts. Above a certain reference altitude $z_0$ the integration is assumed to evaluate to a reference temperature $T_0$ that is then used to initialize or "seed" the integration of the second part starting at $z_0$. Typically, $z_0$ is chosen as the upper boundary of the lidar return profile where the signal is still reliable, and the corresponding $T_0$ is taken from other measurements or models (e.g Duck et al., 2001; Alexander et al., 2011). The condition 'no aerosols present' can be relaxed if inelastic (Raman) scattering
e.g. at nitrogen molecules is used instead of the elastic scattering to derive profiles of relative number density.

In our implementation of the retrieval, the basic preparatory steps prior to the integration are: binning of the photon count data to a 100 m vertical grid and a desired temporal resolution, correction of detector dead-time effects, subtraction of the background which is estimated from photon count profiles between 130 km and 200 km altitude, correction of the two-way Rayleigh extinction, range-correction by multiplication with the range squared, and vertical smoothing to 900 m effective
vertical resolution to improve the signal-to-noise ratio (SNR). After performing these steps, temperatures are computed by integration of the profiles of relative density. This process is repeated independently for each detector channel. The question is now, where do we get the seed value from? We start off with the nightly mean profile of the far-channel which we seed with an approximately co-located SABER (Sounding of the Atmosphere using Broadband Emission Radiometry instrument on the TIMED satellite) profile at typical altitudes 98-108 km. As coincidence criteria we chose a maximum horizontal distance of
1000 km between the mean location of the satellite measurement and the location of CORAL. Because the density profiles from the lower channels overlap with an upper channel, we can then seed the retrieval of the mid-channel temperature profile with a temperature value from the far-channel. In a similar way, both the low-channel and Raman-channel are seeded with a value taken from the mid-channel temperature profile. The altitude where the integration starts and the seed value is taken is determined by the SNR $N^*/\sqrt{N}$ where $N$ is the number of detected photons per 100 m bin and $N^*$ the desired signal
(background subtracted). We define the seed altitude as the maximum altitude with SNR $> 4$ (far-channel) and SNR $> 15$ (all other channels). The criterion SNR $> 4$ at 100 m bin width translates to a relative uncertainty in psuedo-density of 8.3 % in case of the default resolution of 900 m and 6.5 % for 1500 m resolution. For reference, a threshold of 10 % was used by Alexander et al. (2011).

The individual temperature profiles are then merged into a single continuous profile. In order to guarantee a smooth transition
from one profile to another, we compute a weighted average in the overlapping region using the weighting function $w(z) = 0.5 + 0.5 \cos(\pi(z - z_0)/\Delta z)$ within the transition region starting at altitude $z_0$ and vertical extent $\Delta z$. The bottom of the upper of the two profiles is typically chosen as $z_0$ and $\Delta z = 2$ km. Figure 8 shows an example of individually retrieved temperature profiles and overlapping regions. Note the large discrepancy between the Raman temperature profile (channel 4) and the elastic temperature profile (channel 3) at about 20 km altitude, which is caused by stratospheric aerosols. We mitigate the impact of
stratospheric aerosols by transitioning to Raman temperatures below 32 km altitude when merging profiles.

In order to achieve higher temporal resolutions, we implemented the iterative approach sketched in Fig. 12a. After retrieving the nightly mean profile seeded with SABER data, we bin the photon count data to overlapping 120 min wide bins that are offset by 30 min. Temperature profiles are then retrieved from these binned data using seed values taken from the nightly mean profile. This works because the SNR of the 120 min binned photon count profiles is always lower (or equal) to the SNR of
the nightly mean profile for the same altitude. The result of such a coarse temperature retrieval is shown in Fig. 12b. Having

completed the 120 min retrieval, we bin the photon count data to 60 min resolution using 15 min offsets from one bin to the next bin and start the process over again. This time the seed values are taken from the 120 min temperature profiles. In the next iteration, the temporal resolution is increased to 30 min using seed values taken from the 60 min profiles. Iteration by iteration, a pyramid with ever increasing resolution is built up until the desired resolution is reached and at which point the algorithm stops. In Fig. 12b,c we show an example to demonstrates the effect of increasing resolution on temperature profiles. Where the 120 min retrieval reveals only large-scale structures, in this case signatures of an internal gravity wave with a period of ∼6 h and 22 km vertical wavelength, the high-resolution retrieval (Fig. 12c) yields a multitude of fine details including smaller-scale waves. The semidiurnal tide with a period of 12 h also my be present in Fig. 12b,c but is overshadowed by internal gravity waves with much larger amplitudes. Eckermann et al. (2018) show measurements of tides acquired with a predecessor instrument of CORAL above New Zealand with peak amplitudes of ∼6 K at 85-90 km altitude (their Fig. 12).

The implementation of our retrieval also allows increasing the vertical resolution to e.g. 500 m and 300 m in regions where the SNR is sufficient. A high vertical resolution is especially important for retrieving accurately the large vertical temperature gradients induced by large-amplitude waves that are on the verge of becoming convectively unstable. Due to the SNR required, generally, these very high-resolution retrievals are limited to altitudes of 30-70 km.

A common way for estimating the uncertainty of retrieved temperatures $T(z)$ is computing the error propagation in the hydrostatic integration of the density profiles. Here, the assumed errors of the density profiles are the photon count uncertainties $\sqrt{N(z)}$ scaled accordingly. In our implementation we use a different approach. Starting with the photon count profile $N(z)$ and its uncertainty $\sqrt{N(z)}$, we perform a set of 200 Monte Carlo experiments for each profile. In these experiments, the number of detected photons per bin $N$ is replaced with $N + \alpha\sqrt{N}$, with $\alpha$ being random numbers drawn from a Gaussian distribution with a standard deviation of one and zero mean. Then we apply the data reduction steps described above and run the retrieval separately for each of the 200 synthetic photon count profiles. In a last step, the final temperature profile is computed as the mean of all 200 retrieved profiles and its uncertainty is given by the standard deviation.

The Monte Carlo method has the big advantage that all data reduction steps are included in the assessment of temperature uncertainties $\Delta T(z)$ in a completely natural way. That also applies to the initial seed temperatures taken from the SABER profile during the first iteration i.e. retrieval of the nightly mean profile. Of course, the seed temperature is also fraught with uncertainty due to true measurement errors, but also because the SABER measurements may have been acquired up to 1000 km away from the location of CORAL. In order to include the impact of variations in seed temperature, we generate a set of seed profiles of the form $T_{\text{seed}}(z) + \alpha\Delta T(z)$, one for each of the 200 synthetic photon count profiles, with $\alpha$ being again random numbers that are different for each $z$ and have a standard deviation of one and zero mean. In case of the SABER profile $\Delta T(z)$ is assumed constant with a value of 20 K, whereas for subsequent iterations we use the uncertainty of the temperature profile retrieved in the previous iteration. This scheme ensures that the initialization error caused by the SABER profile is passed on to all lidar temperature profiles, resulting in robust uncertainty estimates for retrieved profiles.

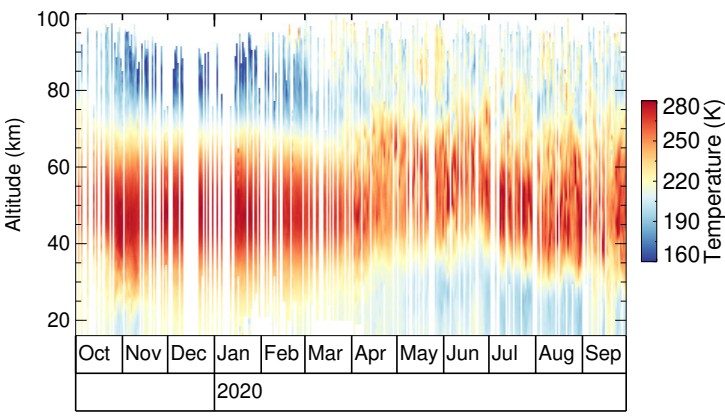

**Figure 13.** Nightly mean temperature profiles acquired by CORAL over a 12-month period in 2019 and 2020.

## 5 Discussion

The CORAL instrument is the first middle atmosphere lidar that is capable of operating fully autonomously for extended periods. By putting computer software in charge and not relying on human operators to start, monitor and stop the instrument, CORAL is destined to acquire atmospheric profiles whenever weather conditions allow for optical soundings. This not only maximizes the data return, but also minimizes potential sampling biases. As demonstrated in Fig. 11, CORAL also operates in marginal weather and records data during short windows with gaps in the cloud layer, an opportunity normally not seized by conventional lidars because of the waste of time of the human operator. We believe that probing the atmosphere as often as possible is critical to capturing the true state of the atmosphere, in particular with regard to atmospheric gravity waves. There is a longstanding and ongoing discussion as to whether lidars in general underestimate wave activity because, being optical instruments, lidars are typically run during stable weather and clear skies conditions. However, for example, it is reasonable to assume that strong forcing of mountain waves occurs in stormy weather conditions, which are often accompanied by clouds. In contrast to conventional lidars, CORAL may operate under these conditions and thus capture strong wave responses not previously observed by other lidars. A study trying to quantify this potential "nice weather" bias is in preparation.

On the other hand, frequent observations—even if they are short—can reveal new insights in the evolution of gravity wave events and the question about their intermittency. Kaifler et al. (2020) detected large-amplitude and long-lasting (several days) mountain waves in CORAL measurements above Rio Grande in southern Argentina. The magnitude of this event would have been certainly heavily underestimated given the coarser sampling of conventional lidars. Ehard et al. (2018) compared CORAL gravity wave measurements acquired above Sodankylä, northern Finland to gravity wave potential energy densities retrieved from the ECMWF IFS. In their study, Ehard et al. showed that IFS forecasted the evolution of wave events reasonably well although wave amplitudes in the upper stratosphere were heavily underestimated. Again, the high cadence of the CORAL measurements turned out to be crucial for this study. A third example highlighting the importance of frequent observations is the work by Kaifler et al. (2015a) who investigated the influences of source conditions on mountain wave penetration into the mesosphere above New Zealand using a predecessor instrument of CORAL. Finally, Kaifler et al. (2017) analyzed CORAL data

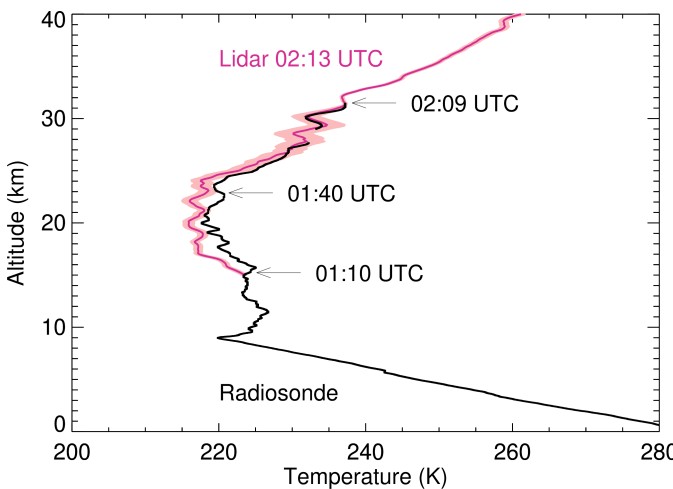

**Figure 14.** Radiosonde and lidar profile acquired on 16 November 2019. The arrows indicate the height of the radiosonde at different times. Reception of the radiosonde signal was briefly lost at 28 km altitude. The shaded area marks the uncertainty of the lidar profile.

for signatures of downward propagating gravity waves. Since these are relatively rare events, collecting as much observations as possible greatly improves the chances for finding cases where the downward propagating waves are not masked by interference with strong upward propagating waves. During a previous campaign in the Bavarian Forrest in Germany CORAL also was able to capture a rare mid-latitude Noctilucent Cloud event (Kaifler et al., 2018).

The above examples clearly demonstrate the scientific value of autonomous lidar systems. However, we also note that site selection plays an important role in the data return of an instrument, as even the most powerful lidars can't operate if there is a constant thick cloud layer above. Based on our experience, sites in the lee of mountains are good places for setting up optical instrumentation. Figure 13 shows nightly mean profiles acquired by CORAL at Rio Grande, southern Argentina, in the lee of the southern Andes. Over the 12-month period CORAL conducted observations on 243 nights, which equals about two out of

three nights or 66.4 %. To our knowledge, no other lidar instrument achieved a similar high cadence over such a long period. Kaifler et al. (2015b) report 74 observations with a daylight capable iron lidar in 2011 (20 %). Li et al. (2018) state 154 nights with sodium lidar observations in 2012-2016 (8 %), Jalali et al. (2018) 519 Rayleigh lidar observations in 1994-2013 (7 %), and Llamedo et al. (2019) 302 Rayleigh lidar observations in 2005-2015 (8 %). The comparison of latter numbers (7-20 %) with CORAL observations (66 %) shows the extraordinariness of the CORAL data set.

The reliability of CORAL measurements is demonstrated in Fig. 14 which shows a comparison between a radiosonde and a lidar temperature profile. The coincident radiosonde sounding was acquired on 16 November 2019 during the Southern Hemisphere Transport, Dynamics and Chemistry (SOUTHTRAC) campaign (Rapp et al., 2020). The radiosonde was launched at Rio Grande airport at ∼1 km distance from the location of CORAL at 00:09 UTC. Almost two hours later, a hole in the cloud layer of ∼35 min duration enabled lidar observations and the retrieval of a temperature profile with 20 min integration

period centered at 02:13 UTC. This period coincides with the upper part of the radiosonde sounding. Both profiles are in nearly

perfect agreement with a mean difference of $< 0.6$ K between 28 km and 33 km altitude. While mean differences increase to $\sim 2.7$ K at the bottom of the lidar profile, we should keep in mind that the radiosonde crossed these altitudes about 1 h earlier and atmospheric conditions have likely changed in the meantime. The lidar measurement was acquired in fully automatic mode demonstrating that the conscan method achieved beam overlap and tracking in less than 10 min after a cold start of the system. Furthermore, the comparison shown in Fig. 14 supports the conclusion that range-dependent defocusing of the telescope is not significant, as otherwise we should see a dramatically diverging lidar temperature profile at altitudes below 20 km (see section S2 in the supporting information).

From an operational point of view, after the completion of the testing phase, CORAL exceeded all expectations. The instrument has been collecting data on a routine basis for the last 13 months without requiring any on-site service, and is still operating normally as of October 2020. That not only demonstrates the robustness of the system, but also proved to be very important for the continuation of the long-term observations given the ongoing travel restrictions due to the COVID-19 pandemic. The performance of the lidar is slowly degrading due to buildup of dust on the telescope mirror and laser turning mirror. However, the lidar system has enough performance reserves to continue observations for another year before cleaning will be necessary. The other limiting factor is the lifetime of the deionization cartridge in the primary cooling loop of the laser, which normally lasts about one year. Accepting a higher risk of failure, based on our experience the laser can be operated with the same cartridge for up to two years. The addition of a conductivity meter for monitoring of the coolant is planned as part of a larger upgrade of the lidar instrument.

Most of the teething problems in the early days of CORAL were caused by or related to software problems. For example, a race condition in the software of one of the two MDMs resulted in the telescope hatch not closing during an approaching rain shower, causing flooding of the telescope compartment. In another incident, a configuration error prevented the successful transfer of authority from one MDM to the other when the first MDM was disabled by a faulty power supply. As a result, environmental control was lost, and freezing of the primary cooling loop of the laser lead to permanent damage and made replacement necessary. These examples show that quality assurance in software development is of similar criticality for the success of an instrument like CORAL as the design of the hardware. Whereas most of the hardware comprises off-the-shelf components, the software that empowers CORAL to make autonomous observations is unique. The same applies to the beam tracking system described in section 2.3.1, which to our knowledge is the first application of the conical scan method to middle atmosphere lidars. Again, the hardware is rather simple and straight forward to implement in lidars, and it is the software that represents a major advancement in technology.

Based on the above examples, we argue that, in the past, the software was an often overlooked and underappreciated aspect in the development of lidar technologies. That even may have been a natural consequence given that most lidars are built by physicists and engineers who usually are no trained software developers. In order to facilitate the development of new technologies including advances in hardware and software, we propose to include trained software developers in lidar teams. Furthermore, we encourage the lidar community to share algorithms and software. Based on our experience with CORAL, it takes considerable efforts and resources to develop software and test it. In our opinion it is a waste of resources if every group has to start from scratch developing more or less the same set of tools.

## 6 Conclusions

This work has described a new autonomous middle atmosphere temperature lidar that is capable of performing fully automatic observations over extended periods, potentially years. This capability represents a major advancement over conventional lidars that are operated only during campaigns or during certain days per week. Not only does the automatic system result in signifi-
565 cantly reduced operating costs as no personnel is needed to run the lidar, but the high cadence of the observations also enables new scientific studies. Thus, the CORAL system is a valuable tool to the scientific community and its success may prompt the development and installation of a whole new class of middle atmosphere lidars that can be used for a broad range of scientific studies e.g. atmospheric gravity wave research, climate monitoring, and satellite data validation. In order to facilitate scientific progress and seed the development of CORAL-type lidars, we will make the CORAL software available to the community.

*Code availability.* The CORAL software is available to the community upon request. It may be placed in a public software repository in future.

*Data availability.* Quicklook plots and real-time status information are available on the instrument web site http://extern05.pa.op.dlr.de/coral.

*Author contributions.* BK designed the CORAL system and wrote the manuscript. NK and BK developed the CORAL software. NK contributed Figures 10 and 11.

*Competing interests.* The authors declare that they have no conflict of interest.

*Acknowledgements.* The authors thank Christian Büdenbender (laser modification) and Philipp Roßi (electronics) for their support during construction and maintenance of CORAL, Jose Luis Hormaechea for resolving technical issues on-site, and Sonja Gisinger for the radiosonde data. Robert Reichert supported the installation and maintenance during on-site visits. Access to ECMWF forecast data was granted through the special project 'Vertical Propagation of Internal Gravity Waves'. The deployment of CORAL to South America and the first year of
580 operations were financed under the umbrella of ARISE2 (http://ARISE-project.eu), which received funding from the European Commission's H2020 program (grant agreement 653980). The initial development of CORAL was funded through the postdoctoral program of the Helmholtz Association (project PD-206).

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
