# Peer review of "A Compact Rayleigh Autonomous Lidar (CORAL) for the middle atmosphere"

_Atmospheric Measurement Techniques, 2020_

## Referee Comment (RC1) · Anonymous Referee #1 · 23 Nov 2020

Review of 'A Compact Rayleigh Autonomous Lidar (CORAL) for the middle atmosphere' by Kaifler and Kaifler

General comments

This paper describes a novel autonomous lidar for temperature retrievals in the middle atmosphere that will assist studies of atmospheric processes on a wide variety of timescales. The description of the architecture and methods employed conveys that the instrument is an advance for the field and is able to enhance opportunities for data collection through automation. This description will be of interest to other workers in the field of middle atmosphere lidar by providing a practical basis on which to tackle the vexed issue of maximising the scientific return from high power-aperture product lidars.

I recommend the work for publication after the authors address the specific and technical comments below. These comments are made from the perspective of someone familiar with the practicalities of middle atmosphere lidar in order to engender into the manuscript additional information to more adequately describe the system and methods employed. The standard of presentation and grammar of the manuscript is very good, the figures and tables are appropriate and concise, and the methods used and their description are sound. Overall, I regard that addressing the following comments will constitute a minor revision of the manuscript as significant rewriting and figure changes are not needed.

Specific comments

Abstract: The third sentence indicates that the first studies with CORAL show the impact of a strong gravity wave event on stratospheric circulation. I initially took this to mean that this paper included this analysis, but after re-reading the Discussion section I see that this is done in earlier papers. I suggest that the 3rd sentence of the abstract be modified to make this clearer (e.g 'First studies using CORAL data have shown for example…).

L87. Please quantify the typical shot-to-shot beam pointing stability of the laser beam after divergence and comment on how this compares with the field of view of the telescope and the divergence of the transmitted beam.

L102. What is the blocking level of your Raman filter at 532 nm?

Section 2.2. Given that the arrangement of the receiver and transmitter are bistatic and that the field of view of the telescope is small (370 μrad), the effect of any significant change in focus will depend on how the response function of the field of view changes and the response function of the field of view. How is the field of view of the telescope practically determined (theoretically or measured)? Please comment on the stability of the telescope focus (e.g. the significance of changes due to thermal effects) and how you determine best focus. Please also comment on if and how you can determine the significance of any range-dependent effect on the retrieved temperature profiles due to focus change.

L98 (or thereabouts). Please indicate what the expected altitude for full overlap between the fields of view of the transmitter and receiver (presumably below 14 km).

Table 1. What is the quality of the surface of the telescope in terms of RMS surface deviation (in wavelengths) from an ideal paraboloid?

Section 2.3.1. Given that the arrangement of the receiver and transmitter are bistatic, I expect that clouds cause the conscan method to optimise the overlap for lower altitudes. Is that correct and might this cause any issues with data collection? Do you use any data in cases of cloud (even thin cloud)?

Section 2.3.1. While you have a cloud-monitoring camera (discussed later) it would seem that introduction of a pellicle beamsplitter in the telescope before the fiber would allow you to monitor the quality and stability of the received beam (e.g. as done in Innis et al., 2007 - https://doi.org/10.1117/1.2801411). Could you please comment on the usefulness or otherwise of such an arrangement for data quality control.

Figure 8b. What is the range and time separation for the SABER profile relative to the lidar measurement? Why not show a measurement centred on the time of the ECMWF profile, or is the SABER profile more coincident? What is the source of the ECMWF profile (forecast or analysis) and what is the horizontal grid resolution of the ECMWF data and separation from the measurement site?

Section 4. Given that you can be seeding at altitudes of 100 km or above (e.g. Fig. 12b and line 355), do you take into account the change in the mean molecular mass of the air in the MLT region, and if so, how?

Figure 12. Is the seeding altitude at the top of the retrievals in panels (b) and (c) or has allowance been made for convergence of the retrieval?

L352. You are describing your retrieval method in fairly general terms and I would recommend that you are more specific, referencing earlier work as necessary or expanding Section 4 to more fully explain the particular criteria and assumptions that you are using. What are the coincidence criteria that you apply in the use of satellite data for seeding? Do you account for bias in the seed temperature? The averaging kernels of the MLS measurements are coarse compared to SABER in the MLT. When would you use MLS measurements to seed the profiles? How do you determine what effective height to assign to the seeding temperature obtained from MLS given that the vertical resolution of the lidar data is much less than the averaging kernel of the MLS retrieval (8-10 km)? Is the 100-108 km altitude range within the recommended upper limit for scientifically useful MLS data at your site. Is there a suitable MLS profile you can show in Fig. 8?

L360. Can you please indicate what the SNR threshold for the far-channel corresponds to as a relative uncertainty in pseudo-density (e.g. section 1.2 in Wing et al., 2020 - https://doi.org/10.3390/atmos11010075 and Alexander et al., 2011 - https://doi.org/10.1029/2010JD015164 who use 20% and 10% for their seeding threshold, respectively)?

L379. By reference to earlier analysis of CORAL data, please indicate if any significant tidal signatures are present in the MLT data in Fig 12c and what characteristics of the temperature variations allow you to conclude a 5-7 hour period internal gravity wave is present (e.g. downward phase progressions or narrow duty cycle,  perhaps highlighting with dashed lines).

Technical comments

L12. Define ECMWF (done at L280, so will need to adjust at that point too).

L37. CORAL likely needs to be defined (again).

L38. 'atmospheric' rather than atmosphere.

L39. Consider a gender-neutral alternative to manpower (e.g. 'human effort').

L70. Define DLR.

L71. Define DEEPWAVE.

L77. Depending on the style requirement of the journal, may need to spell out 8' as 8-foot. If this is a standard size, perhaps indicate 8-foot ISO container, and define ISO.

L105. photomultiplier (as one word)

L106. interference filter

L421 IFS is spelled out unnecessarily (see L280).

---

## Referee Comment (RC2) · Anonymous Referee #2 · 25 Nov 2020

This study describes a new Rayleigh/Raman lidar dedicated to temperature retrieval. The novelty of the results presented here concern the fully autotomized mode for lidar operation that is presented for the first time. The manuscript is well written and the results about the lidar autonomy is fully demonstrated by the temperature series presented and will be for sure for the future other systems a reference. For this reason, the publication of this manuscript in AMT is valuable. However, I have several restrictions in the present form and I consider that this manuscript requires some critical improvements. The first one concerns the missing references for many aspects while a lot of works has already been done by the research community. Auto-citing needs to be balanced by work performed by the international research community. The second one concerns the technical choices. no alternatives are presented and this is missing to

convinced readers that the choices performed by authors are the results of an optimum scientific choice and to avoid part of the manuscript being a technical note rather than a scientific contribution. Finally, the capabilities of the lidar need to be discuss and mainly regarding the automatization issues. How this mode does not perturb nominal capabilities, and what are the proxy used to check the temperature retrieval quality. No comparisons with other observations are presented. While operators introduce some uncertainties in the quality of the observations, automatic mode main also introduce some bias. How can we check this issue? I am sure that authors have the response to my questions and I think it will be valuable to further document the missing information. For my point of view this manuscript requires significative revisions that can be quickly handle.

**Detailed suggestions**

Line 24 page 1 The comment about the fact that lidar only operate during campaigns is a wrong statement while within the NDACC network routine measurements are performed over many sites around the world. The longest data based is obtained at Observatory of Haute-Provence with more than 40 year of continuous observations. Many publications are related to these long commitments. The only thing true is that these systems do not have fully automatic mode for their operations (some of them have semi-automatic mode with possibility to stop when rain and cloud are coming) and require operators for turning them on and ensuring alignment. For data analyses some NDACC partners have automatic software's to process the data real-time including automatic data cleaning. This statement needs to be modified and a section about the associated works need to be provided while many climatologic works have been performed including trends that were published in international reports for IPCC, ozone assessment or SPARC-WCRP.

Page 2 line 26 Many gravity waves climatology were already performed that need to be cited. One of the first I think was performed by Wilson et al. in late 1980's and early 1990's.
Page 2 line 58 Operators are also required for safety reasons and air traffic control. These issues need to be introduced here and information about the capabilities of CORAL can be discuss and documented later on the manuscript. The issue is; how authorities can have confidence of such system?

Section 2 about lidar description The description here requires some arguments about the choice rather in referencing to past work of the team, to work performed by other teams or in providing evidences through graphic or capability comparisons when technical choices are different from what the scientific community has performed.

Page 6 section about automatic tracking of the laser beam This section is well described and critical for lidar automatization. It requires an introduction explaining the technical choices compare to other methods and also the sinusoidal exploration was not explained. Final capabilities need to be demonstrated. Figure 7 is not so clear for me. Also, the time speed for correct alignment need to be discussed according the sky conditions. This is a critical point while human have a exploration mode that is sometime more efficient while less quantitative. Also the geometry here is coaxial. Is it a requirement or the development described here can be apply to bi-axial systems?

Page 15 section example In this section comparisons with other observations are required. No additional observations will fully validate the full profile except radiosondes. Also, many comparisons between other Rayleigh lidars and satellite instruments have been performed, is CORAL found similar deviations. Authors mentioned MLS and SABER, recent comparisons were published and can used as comparisons. The main point will be about the demonstration that such alignment does not introduce any bias. Comparison with radiosonde and temperature retrieval right after alignment will be a first demonstration.

Within NDACC, lidars ensure their qualification by comparison with a mobile system running by NASA. Many publications have reported about these comparisons. It is out of the scope of this studies but collocated measurements with other lidars will allow to
**convinced the scientific community about the data quality.**

---

## Author Comment (AC1) · 9 Dec 2020

We thank the reviewer for the thorough review of our manuscript and for pointing out shortcomings in the description of the lidar system. When you are intimately familiar with an instrument, it is easy to miss something in the description which may be important to readers who are not so familiar with intricacies of that particular system. Therefore we are very grateful for the review comments and believe that by addressing those the manuscript will be greatly improved. In particular, we want to thank the reviewer for asking about the problems associated with incomplete beam overlap and focusing issues of the telescope. These are often overlooked in literature, our manuscript included. We address these problems as well as other comments in the following and will include relevant information in the revision of the manuscript.

[Figure]

**1  The third sentence indicates that the first studies with CORAL show the impact of a strong gravity wave event on stratospheric circulation. I initially took this to mean that this paper included this analysis, but after re-reading the Discussion section I see that this is done in earlier papers. I suggest that the 3rd sentence of the abstract be modified to make this clearer (e.g 'First studies using CORAL data have shown for example...).**

We agree, the sentence is indeed misleading. We will modify it in the revision to make it clear that it refers to an earlier study.

**2  Please quantify the typical shot-to-shot beam pointing stability of the laser beam after divergence and comment on how this compares with the field of view of the telescope and the divergence of the transmitted beam.**

We haven't measured the shot-to-shot beam pointing stability of this particular laser. The data sheet provided by the manufacturer lists $< \pm 50$ μrad as pointing stability. Expanding the beam reduces the pointing jitter to $< \pm 25$ μrad or 13.9 % of the receiving telescope FOV (361 μrad full angle; note that we falsely stated 370 μrad in the manuscript). Measurements performed with other lasers of the same manufacturer and type suggest a significantly smaller pointing jitter in the order of 25 μrad. Assuming a Gaussian distribution, the pointing jitter leads to an effective increase in the divergence of the laser beam when averaging over multiple laser shots. Assuming the upper limit of $50$ μrad full angle, the effective beam divergence including the pointing jitter is $220$ μrad, or 195 μrad in case of the more realistic estimate of 25 μrad pointing jitter.

**3 What is the blocking level of your Raman filter at 532 nm**

The specification for the out-of-band blocking of the Raman filter is optical depth (OD) >6. The dichroic mirror which separates the elastic scattering and Raman scattering has a transmission of 1.2 % at 532 nm wavelength. Both optics combined result in a blocking level of OD $\sim 8$ at 532 nm.

**4 Given that the arrangement of the receiver and transmitter are bistatic and that the field of view of the telescope is small (370 µrad), the effect of any significant change in focus will depend on how the response function of the field of view changes and the response function of the field of view. How is the field of view of the telescope practically determined (theoretically or measured)? Please comment on the stability of the telescope focus (e.g. the significance of changes due to thermal effects)and how you determine best focus. Please also comment on if and how you can determine the significance of any range-dependent effect on the retrieved temperature profiles due to focus change.**

The field of view of the telescope is given by the diameter of the fiber core a=550 µm and the focal length of the mirror $f = 2.4\,d$ ($d = 0.635$ m) (see Fig. 1):

$$\text{FOV} = \frac{5.5 \times 10^{-4} \text{ m}}{2.4 \times 0.635 \text{ m}} = 361 \text{ µrad} \tag{1}$$

Note that we falsely stated 370 µrad in the manuscript.

**4.1 Effect of thermal expansion**

Thermal expansion and contraction of the telescope, most importantly in the z-axis or focus direction, is minimized by using carbon fiber tubes as structural member and spacer between the mirror cell and the telescope spider. The coefficient of thermal expansion of carbon fibers is $-0.1 \times 10^{-6}$ K$^{-1}$ along the fiber and $21 \times 10^{-6}$ m K$^{-1}$ across. The tubes we used have 90 % of the fibers aligned along the tube and 10 % across for increased torsion stability. Thus, we assume an average coefficient of $2.09 \times 10^{-6}$ m K$^{-1}$. Given the focal length of 1.524 m, a change in temperature of 50 K causes a focus shift of 160 µm.

We can estimate the impact of the temperature induced focus shift on the coupling efficiency as follows. First we assume that the facet of the fiber with core diameter $a$ is fully and equally illuminated by the telescope mirror. The half angle of the illuminating light cone is then given by

$$\tan \beta = \frac{d}{2\,f}. \tag{2}$$

Next we calculate the fraction the diameter of the illuminated disk increases when the disk, previously at the position of the fiber end, is moved downward by $b = \alpha f$ where $\alpha = 2.09 \times 10^{-6}$ K$^{-1}$ is the coefficient of thermal expansion:

$$\frac{a'}{a} = \frac{(2\,x + b)\tan \beta}{2\,x \tan \beta} = 1 + \frac{b}{2x}. \tag{3}$$

Looking at Fig. 1, the angle $\beta$ can also be expressed as

$$\tan \beta = \frac{x}{2\,a}. \tag{4}$$

Eliminating $\tan \beta$ in this equation using relation (2) and substituting the result into relation (3) yields

$$\frac{a'}{a} = 1 + \frac{b\,d}{2af} \tag{5}$$

Substituting the thermal expansion $b = \alpha f$ into this equation results in

$$\frac{a'}{a} = 1 + \frac{\alpha d}{2a} = 1 + 1.21 \times 10^{-3} \text{ K}^{-1}. \tag{6}$$

Hence, we expect a 6.0 % increase in the diameter of the illuminated disk for a 50 K change in temperature. However, the fraction of the light coupled into the fiber core decreases with the diameter squared i.e. a 6.0 % increase in diameter results in a net loss of only 0.36 %. In reality, as will be shown in the next section, the signal loss is much smaller because a) the laser beam underfills the FOV of the telescope and b) more energy is concentrated in the center because of the Gaussian beam profile. We conclude that thermal expansion of the telescope has a negligible influence on the performance of the lidar.

When setting up the lidar, we scan the vicinity of the expected focus position by slowly moving up and down the fiber end using the focus motor. Because the motor is equipped with encoders, we can record the strength of the detected lidar signal in the altitude range 40-50 km as function of position. Repeated scans are performed and measurements averaged in order to reduce the effect of potential changes in atmospheric transmission during the scans. We find the optimum focus position by fitting the position of the signal maximum.

**4.2 Range dependent effects**

In order to evaluate range dependent effects, we look at the function $P(z)$ which describes the fraction of the collected scattered light with $1/e^2$ beam radius $w$ passing through an aperture with radius $r$ as function of altitude $z$:

$$P(z) = 1 - \exp\left(-\frac{2r^2}{w(z)^2}\right) \tag{7}$$

As aperture we define the core of the optical fiber and set $r = 0.5\,a$. The radius of the beam of scattered laser light at the location of the aperture is then given by

$$w = \frac{\epsilon}{\text{FOV}} \frac{a'}{2} \tag{8}$$

where $\epsilon$ is the full-angle divergence of the imaged laser beam taken at $1/e^2$ points and FOV is the full-angle of the telescope FOV. Because the beam divergence of 170 µrad was measured at $1/e$ points we have to scale the value by 1.41. In addition, we add the effective broadening due to the pointing jitter of the laser (25 µrad) and broadening due to the point spread function of the telescope (30 µrad since the spot size of 60 µrad refers to the $2\sigma$-value) and arrive at the beam radius of $0.408\,a'$ at the facet of the fiber.

Even if the laser beam is completely within the FOV of the telescope (full geometric overlap is achieved at 4.82 km altitude), signal may be lost due to defocusing of the telescope at low altitudes. To estimate this range dependent signal loss, we first calculate the focus shift associated with moving an object (scattered laser light) from infinity to the near vicinity of the telescope and then evaluate how this focus shift affects the coupling efficiency of telescope and fiber in a similar way as was done when analyzing the effect of thermal expansion.

Looking at Fig. 1, the angular change of the object rays is given by

$$\gamma = \frac{d}{2z}. \tag{9}$$

with $z$ being the altitude. Here we have used the small-angle approximation $\tan \Theta \approx \Theta$. Moving the object closer i.e. increasing $\gamma$ results in an increase of the focal length of the telescope by $\delta$. The new angle

$$\beta' = \beta - \gamma \tag{10}$$

is defined by relating the new focal length to the diameter of the telescope mirror as given by

$$\tan \beta' = \frac{d}{2\left(f + \delta\right)}. \tag{11}$$

Solving this equation for $\delta$ and substituting relations (2), (9) and (10) yields

$$\delta = \frac{d}{2 \tan\left(\tan^{-1}\left(\frac{d}{2f}\right) - \frac{d}{2z}\right)}. \tag{12}$$

Now we solve Eqn. (5) for $a'$, replace $b$ with the focus shift $\delta$, and together with Eqn. (8) substitute the result into Eqn. (7):

$$P\left(z\right) = 1 - \exp\left(-2\frac{\text{FOV}^2}{\epsilon^2}\frac{1}{\left(1 + \frac{\delta d}{2af}\right)^2}\right) \tag{13}$$

Note that the exponential depends on the inverse of the filling factor $\epsilon\,\text{FOV}^{-1}$ that describes the ratio of effective laser beam divergence and the telescope FOV, and the scaling term that takes into account the broadening of the imaged laser spot caused by the focus shift. Fig. 2 shows the altitude dependence of $P$ and also includes two additional gray lines, the upper line marking the altitude (45 km) for which the focus position position was experimentally determined as described above, and the lower line marking our threshold of 15 km altitude. The difference in the encircled energy between the two gray lines is 1.35 %. Assuming a mean stratospheric temperature of 220km, this difference translates into a temperature error at 15 km altitude of 2.7 K. Below, the temperature error rapidly increases, as indicated by the curve in Fig. 2. For altitudes above 30 km predicted errors are below 0.3 %. The actual errors are likely smaller as suggested by a detailed comparison study which is currently in preparation

for publication and analyzes CORAL temperature profiles and ECMWF profiles in the lower stratosphere.

So far we do not attempt to correct this range dependent effect in our implementation of the temperature retrieval because, in addition to the above theoretical considerations, a more thorough experimental study of the overlap function is needed.

**5  Please indicate what the expected altitude for full overlap between the fields of view of the transmitter and receiver (presumably below 14 km).**

The altitude at which full overlap is achieved is defined as the point where the marginal rays of the laser beam and the telescope intersect. Given the horizontal separation of 0.4 m of the laser beam and telescope axis, the effective beam divergence of 195 µrad and the telescope FOV of 361 µrad, the intersection occurs at the altitude of 4.82 km. Full overlap is obtained for altitudes above.

**6  What is the quality of the surface of the telescope in terms of RMS surface deviation (in wavelengths) from an ideal paraboloid?**

We do not have any measurements of the surface quality of the telescope mirror.

**7  Given that the arrangement of the receiver and transmitter are bistatic, I ex-
pect that clouds cause the conscan method to optimise the overlap for lower
altitudes. Is that correct and might this cause any issues with data collec-
tion? Do you use any data in cases of cloud (even thin cloud)?**

Actually, the conscan method appears to be fairly robust with regard to clouds. One
major benefit over the more widely used imaging method described in Innis et al. (2007)
is that the lidar return signal is analyzed well above potential cloud layers. In our
implementation we evaluate the signal between 45 km and 55 km altitude. Though
clouds passing through the lidar beam modulate the strength of the lidar return signal
and thus impact the demodulation of the conscan error signal, this does not change
the altitude for which the beam overlap is optimized. For that reason, we trust our lidar
measurements and use the data in case of thin clouds. Thick clouds cause the lidar
return signal to drop below a threshold and raw data profiles, we use 10 s intervals for
quality checks, are automatically discarded in the temperature retrieval.

We also use quality checks in our conscan implementation. For example, we evaluate
the $\chi^2$-value retrieved from the fit (shown in Fig. 3). A large $\chi^2$-value, we use 300 as
a threshold, indicates that the conscan signal could not be properly demodulated and
retrieved estimates of phase and amplitude are not reliable. In this case the conscan
cycle is aborted and the beam pointing not updated. In cloudy conditions as many as 9
out of 10 conscans may fail in that way, but the succeeding conscans are still sufficient
for beam tracking, as thermal drifts happen on relatively large time scales. If more than
10 successive conscans fail, subsequent intervals are marked in the raw data files as
potentially having incomplete beam overlap. Moreover, beam pointing increments and
conscan parameters are stored in raw data files for offline analysis. As evident from
Fig. 3 only few conscan measurements exceed the the threshold of $\chi^2 > 300$ even
though the lidar return signal was strongly impacted by clouds in the period 4-5 UTC.
Note that we did not normalize the conscan signal prior to fitting the data. Hence, the

average $\chi^2$-value is about 20 instead of unity even for strong conscan signals.

**8 While you have a cloud-monitoring camera (discussed later) it would seem that introduction of a pellicle beamsplitter in the telescope before the fiber would allow you to monitor the quality and stability of the received beam (e.g. as done in Innis et al., 2007 -https://doi.org/10.1117/1.2801411). Could you please comment on the usefulness or otherwise of such an arrangement for data quality control.**

We choose not to use a beam tracking camera as it was done in Innis et al. (2007) for two main reasons. First, pellicle beam splitters are extremely fragile and are likely not to survive for long periods in a humid environment. CORAL is located close to the coast and we operate it in marginal weather conditions such as thin fog. A plate beam splitter may solve this problem, but complicates the optical setup. Second, the addition of a beam splitter increases signal losses i.e. less light is available to the receiver. With CORAL we wanted to demonstrate that even small lidars (physical size and laser power) can produce good measurements in the mesosphere if they are optimized for low signal losses.

For some time we used a beam tracking camera looking through a separate Galilei telescope (100 mm aperture) bolted to the main telescope. We found that the error signal determined from the camera was less reliable in (partially) cloudy conditions as compared to the conscan method. We never used the camera signal to assess the data quality because we are confident in the performance of the conscan method.

**9  What is the range and time separation for the SABER profile relative to the lidar measurement? Why not show a measurement centred on the time of the ECMWF profile, or is the SABER profile more coincident? What is the source of the ECMWF profile (forecast or analysis) and what is the horizontal grid resolution of the ECMWF data and separation from the measurement site?**

The closest SABER profile was acquired at 525 km mean distance to the west of CORAL at 05:38 UTC. We initially chose to start the lidar integration period at the beginning of the hour, but agree that centering it at the time of the SABER profile is more appropriate. Fig. 4 shows the temperature profiles for the updated time period.

We use operational ECMWF forecast data with a grid resolution of $0.125°$. The grid point closest to the location of CORAL ($53.785°$ S, $67.752°$ W) is $53.75°$ S, $67.75°$ W with a distance of 3.9 km.

**10  Given that you can be seeding at altitudes of 100 km or above (e.g. Fig. 12b and line 355), do you take into account the change in the mean molecular mass of the air in the MLT region, and if so, how?**

No, we don't take any change in mean molecular mass into account. This is a shortcoming of our retrieval and will be addressed in the future.

**11  Is the seeding altitude at the top of the retrievals in panels (b) and (c) or has allowance been made for convergence of the retrieval?**

The profiles shown in Fig. 12b,c of the manuscript start 2 km below the seeding altitude. Normally you would integrate at least one density scale height to allow for the retrieved

temperature profile to converge. However, as our group mainly focuses on gravity wave studies and thus predominantly looks at perturbations with mean background temperature profiles subtracted, reaching higher altitudes is of higher importance than absolute convergence. Please note that the 2 km refers to the retrieved profile of the current level within the retrieval pyramid. For example, the 20 min profiles shown in Fig. 12c. start at $\sim 90$ km which is more than one density scale height below the altitude where the nightly mean profile was seeded with the SABER profile. Hence, we can assume that the seeding temperatures we used for seeding the 20 min temperature profiles 2 km above are already close to the true atmospheric temperature, allowing for rapid convergence.

**12   You are describing your retrieval method in fairly general terms and I would recommend that you are more specific, referencing earlier works necessary or expanding Section 4 to more fully explain the particular criteria and assumptions that you are using. What are the coincidence criteria that you apply in the use of satellite data for seeding? Do you account for bias in the seed temperature? The averaging kernels of the MLS measurements are coarse compared to SABER in the MLT. When would you use MLS measurements to seed the profiles? How do you determine what effective height to assign to the seeding temperature obtained from MLS given that the vertical resolution of the lidar data is much less than the averaging kernel of the MLS retrieval (8-10 km)? Is the 100-108 km altitude range within the recommended upper limit for scientifically useful MLS data at your site. Is there a suitable MLS profile you can show in Fig. 8?**

We assumed that the hydrostatic temperature retrieval is sufficiently described in literature and discussed only the non-standard parts specific to our implementation. However, we agree that this section should be expanded to benefit a larger audience.

As coincidence criteria we chose a maximum horizontal distance of 1000 km between the mean location of the satellite measurement and our lidar. Normally, we used SABER temperature profiles for seeding, but for rare cases when SABER data are not available, e.g. when there were problems with the instrument, we fall back to using MLS profiles. MLS seed profiles account for $\sim 4$ % of all seed profiles. Given the relatively large averaging kernel and temperature biases of MLS profiles in the lower thermosphere, a better approach may be the use of seed profiles derived from a SABER climatology instead of MLS data when SABER profiles are not available. In any event, we agree that substituting MLS profiles for missing SABER profiles without careful analysis may have been naive and we are grateful for the hints provided by the reviewer.

Currently we do not account for any biases satellite temperature profiles may have.

**13 Can you please indicate what the SNR threshold for the far-channel corresponds to as a relative uncertainty in pseudo-density (e.g. section 1.2 in Wing et al., 2020 -https://doi.org/10.3390/atmos11010075 and Alexander et al., 2011 -https://doi.org/10.1029/2010JD015164 who use 20 % and 10 % for their seeding threshold, respectively)?**

Our criterion of SNR $> 4$ in relation to 100 m bin width translates to a relative uncertainty in pseudo-density of 8.3 % for the default vertical resolution of 900 m and 6.5 % for 1500 m resolution.

**14** **By reference to earlier analysis of CORAL data, please indicate if any significant tidal signatures are present in the MLT data in Fig 12c and what characteristics of the temperature variations allow you to conclude a 5-7 hour period internal gravity wave is present (e.g. downward phase progressions or narrow duty cycle, perhaps highlighting with dashed lines).**

We tried to investigate tides using the CORAL data set but had very limited success. The two main problems are that in winter temperature variability is dominated by mountain waves, and in summer, when gravity wave amplitudes are low, measurements are not long enough to capture tides. Measurements in spring indicate amplitudes of the semi-diurnal tide of <5 K up to 80 km altitude. Our amplitude estimates are consistent with amplitudes reported by Lübken et al. (2011), though we note these measurements were obtained in summer and at a higher latitude (69°S). Eckermann et al. (01 Aug. 2018) show measurements of tides acquired with the TELMA predecessor of CORAL above New Zealand with peak amplitudes of $\sim 6$ K at 85-90 K (their Fig. 12). Tides with similar amplitudes may be present in Fig. 5, but are certainly overshadowed by internal gravity waves which can have much larger amplitudes of 10-25 K in the upper mesosphere.

The dominating features in the mesosphere in Fig. 5 are two warm anomalies that propagate downward in time and are marked by dashed lines. Based on the observed period of $\sim 5$ h and 22 km vertical wavelength we conclude that these features are manifestations of internal upward propagating gravity waves.

**References**

Eckermann, S. D., Ma, J., Hoppel, K. W., Kuhl, D. D., Allen, D. R., Doyle, J. A., Viner, K. C., Ruston, B. C., Baker, N. L., Swadley, S. D., Whitcomb, T. R., Reynolds, C. A., Xu, L., Kaifler, N., Kaifler, B., Reid, I. M., Murphy, D. J., and Love, P. T.: High-Altitude

(0?100 km) Global Atmospheric Reanalysis System: Description and Application to the 2014 Austral Winter of the Deep Propagating Gravity Wave Experiment (DEEPWAVE), Monthly Weather Review, 146, 2639 – 2666, https://doi.org/10.1175/MWR-D-17-0386.1, https://journals.ametsoc.org/view/journals/mwre/146/8/mwr-d-17-0386.1.xml, 01 Aug. 2018.

Innis, J. L., Cunningham, A. P., Graham, A. D., and Klekociuk, A. R.: Automatically guiding a telescope to a laser beam on a biaxial antarctic light detection and ranging system, Optical Engineering, 46, 1 – 8, https://doi.org/10.1117/1.2801411, https://doi.org/10.1117/1.2801411, 2007.

Lübken, F.-J., Höffner, J., Viehl, T. P., Kaifler, B., and Morris, R. J.: First measurements of thermal tides in the summer mesopause region at Antarctic latitudes, Geophysical Research Letters, 38, https://doi.org/https://doi.org/10.1029/2011GL050045, https://agupubs.onlinelibrary.wiley.com/doi/abs/10.1029/2011GL050045, 2011.

δ   Focal plane

x   Facet of fiber core

a

b

a'

β

β

β

β'

f

γ

γ

Parabolic mirror

d

**Fig. 1.** Sketch showing the geometry of rays originating from infinity (blue) and near distance (red) and reflected by the telescope mirror.

**Fig. 2.** Fraction of the energy passing through the aperture of the fiber core as function of altitude. The gray lines mark the focusing altitude 45 km and the threshold altitude 15 km.

[Figure]

**Fig. 3.** Performance of the conscan system during the measurement on 3 November 2019, 4-8 UTC. (a) Scan mirror angles, (b) lidar return signal integrated between 45 km and 55 km altitude in units

[Figure]

**Fig. 4.** (a) Photon count profiles acquired on 3 November 2019 between 05:20 and 05:50 UTC and binned to 100 m vertical resolution, and (b) retrieved temperature profiles with 900 m vertical resolution.

(c)

Fig. 5. Retrieved temperature profiles (original Fig. 12c in the manuscript) with gravity wave phase lines indicated by dashed lines.

---

## Author Comment (AC2) · 15 Dec 2020

We thank the reviewer for the in-depth review of our manuscript. There are several aspects which we missed and which need more careful explanation to make our intentions clear and our description concise. We address the comments below and will add missing information in the revision of the manuscript.

[Figure]

**1  However, I have several restrictions in the present form and I consider that this manuscript requires some critical improvements. The first one concerns the missing references for many aspects while a lot of works has already been done by the research community. Auto-citing needs to be balanced by work performed by the international research community. The second one concerns the technical choices. No alternatives are presented and this is missing to convinced readers that the choices performed by authors are the results of an optimum scientific choice and to avoid part of the manuscript being a technical note rather than a scientific contribution.**

It is our intention to describe our system and convey its advantages. Discussing all developments which happened in the past would significantly change the focus of the manuscript and likely turn it into a review style paper. The latter is not our aim nor do we feel qualified to write such a paper since our experience with lidar systems covers only the last decade.

We agree that we should discuss alternative methods for beam tracking since the conscan method is a new concept in mesospheric lidars, and we should point out the rationale for using a bistatic transmitter/receiver configuration. Apart from those modifications, we believe our lidar setup is fairly standard for contemporary mesospheric Rayleigh lidars. It is our opinion that by now e.g. the benefits of using diode-pumped lasers are widely recognized within the community and no explanations are needed any more. It was our desire to keep the manuscript short and focused rather than discussing technological choices which we think are nowadays obvious.

**2** **Finally, the capabilities of the lidar need to be discuss and mainly regarding the automatization issues. How this mode does not perturb nominal capabilities, and what are the proxy used to check the temperature retrieval quality. No comparisons with other observations are presented. While operators introduce some uncertainties in the quality of the observations, automatic mode main also introduce some bias. How can we check this issue?**

We are not entirely sure what the reviewer means by "nominal capabilities". If manual operation is meant, like it is done with non-automatic systems, then this mode of operation is still possible with CORAL. The autocontrol software can be stopped and control switched to remote control by an operator at any time. Moreover, manual operation by throwing switches inside the container is also possible.

In the manuscript we show comparisons with approximately co-located SABER measurements and ECMWF profiles (see Fig. 8). Ehard et al. (2018) present a detailed comparison of lidar temperature profiles and ECMWF data which reveal an almost perfect agreement up to ∼50 km where ECMWF starts to diverge (see their Figure 2). We agree that the section discussing the temperature retrieval is rather short and may need expansion, in particular with regard to assumptions made in the retrieval and with regard to quality checks of the data.

We are not sure what kind of biases an automatic measurement mode should introduce. The automatic measurements are exactly the same as manually controlled measurements with regard to data acquisition. Automatic measurements allow regular probing of the atmosphere at marginal weather conditions, conditions which are usually avoided when operating lidars manually. That may lead to, for example, enhanced gravity wave potential energy densities in those measurements because stormy conditions are thought to favour excitation of mountain waves. We are in the process of studying this potential effect in detail. Actually, within the lidar community there is a long standing discussion whether lidar measurements are biased towards fair weather atmospheric conditions. Following that discussion, our automatic measurements may be a path forward to remove that potential bias in measurements. Thus, automatic measurements may actually remove a potential bias rather than introduce one. The study is still work in progress and preliminary results indicate no bias in mean temperatures. In particular, dividing our measurement data into two subsets comprising of long and short measurements, the latter being usually acquired during marginal weather conditions, does not result in any significant deviations in monthly mean temperatures.

**3  Line 24 page 1: The comment about the fact that lidar only operate during campaigns is a wrong statement while within the NDACC network routine measurements are performed over many sites around the world. The longest data based is obtained at Observatory of Haute-Provence with more than 40 year of continuous observations. Many publications are related to these long commitments. The only thing true is that these systems do not have fully automatic mode for their operations (some of them have semi-automatic mode with possibility to stop when rain and cloud are coming)and require operators for turning them on and ensuring alignment. For data analyses some NDACC partners have automatic software's to process the data real-time including automatic data cleaning. This statement needs to be modified and a section about the associated works need to be provided while many climatologic works have been performed including trends that were published in international reports for IPCC, ozone assessment or SPARC-WCRP.**

We agree that using the word "campaign" may have been misleading. What we meant is that these long-term operated lidar stations generally take routine measurements only during certain days per week when the weather forecast is favorable. For example, at the ALOMAR observatory which with the authors have some experience, no lidar observations are conducted at weekends outside of special campaign periods. The

quasi periodic but sparse observations may be sufficient for producing climatologies and estimating trends as mentioned by the reviewer, but clearly are not adequate for studies requiring dense sampling e.g. the investigation of the temporal evolution of gravity wave events. We will change this statement in the revision of the manuscript to make that clear.

**4  Page 2 line 26: Many gravity waves climatology were already performed that need to be cited. One of the first I think was performed by Wilson et al. in late 1980's and early 1990's.**

We agree that there are many lidar-based gravity wave studies in literature e.g. Wilson et al. (1991); Sivakumar et al. (2006); Rauthe et al. (2008); Li et al. (2010); Mzé et al. (2014); Kaifler et al. (2015). However, non of these studies document the detailed temporal evolution of a mountain wave event lasting several days. It is our point here that automatic instruments like CORAL allow the investigation of the temporal evolution of singular events, which, to the knowledge of the authors, is not possible based on other pre-existing lidar data sets because of the much lower measurement cadence. It was not our intention to indicate that only CORAL-type lidars can produce gravity wave climatologies.

[Figure]

**5  Page 2 line 58: Operators are also required for safety reasons and air traffic control. These issues need to be introduced here and information about the capabilities of CORAL can be discuss and documented later on the manuscript. The issue is; how authorities can have confidence of such system?**

That is indeed an important topic that we neglected in our manuscript. At the southern tip of South America air traffic is so low that no particular safety measures are requested by the authorities in Argentina. We use an Automatic Dependent Surveillance–Broadcast (ADS-B) receiver for receiving aircraft position information. This information is used by the autocontrol software of CORAL to automatically shut down the laser when an aircraft enters a circle of 800 m radius centered at the location of CORAL. Measurements are resumed when the aircraft leaves the circle. However, within the three years of operation, no single aircraft was detected within the critical zone. At the DLR site near Munich we use a restricted area that is closed for air traffic during lidar operations, and during the Deep Propagating Gravity Wave Experiment (DEEPWAVE, see Fritts et al. (2016)) there was again so little air traffic above Lauder, New Zealand, that no special safety measures were required by local authorities.

For other places we are contemplating the use of a radar system in combination with an optical system for detecting approaching aircraft. The output of the radar would be wired directly to the interlock of the laser to bypass any potential software failures. Still, proving the reliability of such a system is a major difficulty for getting approval by local authorities, and operation of a lidar at locations with heavy air traffic may be even denied regardless of any safety measures. In that regard, the density of expected air traffic may be another relevant factor for setting up a lidar at a particular location.

[Figure]

**6  Section 2 about lidar description. The description here requires some ar-
guments about the choice rather in referencing to past work of the team, to
work performed by other teams or in providing evidences through graphic or
capability comparisons when technical choices are different from what the
scientific community has performed.**

The biggest departure from "classical" lidar setups is probably the use of the conscan
method for beam tracking. The reason for preferring this method over other methods
was missing in the manuscript and is discussed in the following section. We will add
this information in the revision of the manuscript. Apart from conscan, the CORAL
lidar setup is straight forward and similar to setups used in other lidar systems. Mi-
nor changes include the selection of a bistatic transmitter/receiver configuration which
eliminates two bending mirrors needed for a coaxial configuration, and consolidation of
the lidar electronics in a single custom-developed electronics box.

**7  Page 6 section about automatic tracking of the laser beam. This section is
well de-scribed and critical for lidar automatization. It requires an introduc-
tion explaining the technical choices compare to other methods and also
the sinusoidal exploration was not explained. Final capabilities need to be
demonstrated. Figure 7 is not so clear for me. Also, the time speed for cor-
rect alignment need to be discussed according the sky conditions. This is
a critical point while human have a exploration mode that is sometime more
efficient while less quantitative. Also the geometry here is coaxial. Is it a
requirement or the development described here can be apply to bi-axial sys-
tems?**

The authors are familiar with two other methods for beam alignment. Manual align-
ment involves moving the laser beam by hand using motorized beam folding mirrors

in the transmitter optical path. The operator commands the motors while watching the strength of the lidar return signal at a certain altitude and tries to find the optimum position where the signal maximizes. This works well when the sky is clear and tropospheric transmission does not change in time, but is absolutely impossible during conditions of e.g. broken clouds. An automatic autoguiding system is described by Innis et al. (2007). In their setup, a camera looks through the telescope via a pellicle beam splitter and images the laser beam at a certain altitude. Images are analyzed and the position of the laser beam is computed. This position information is then used to compute an error signal relative to a target position, and the error signal is subsequently fed to motorized actuators in order to neutralize any deviation from the target. We have used such a system in the beginning but removed it when it became clear that the conscan method performed much better in marginal weather conditions. One of the main problems with the camera based autoguiding system was thin cloud layers passing through the altitude where the beam is imaged. Because Mie scattering within the clouds is much stronger than Rayleigh scattering, the autoguiding algorithm tended to track features within the clouds rather than the actual beam position. This caused a misalignment of the laser beam and, given our rather small FOV, resulted in incomplete overlap and thus unusable data.

The conscan method does not suffer from this problem because the lidar return signal is evaluated at altitudes well above potential cloud layers. We have added additional panels to Fig. 7 (shown here in Fig. 1) to show the phase, amplitude, and $\chi^2$-signal retrieved from conscan measurements. As pointed out in our reply to Reviewer Comment #1, the conscan signal is impacted by changes in tropospheric transmission caused by clouds and demodulation of the conscan signal may fail sometimes. This condition is detected by the conscan algorithm and the current conscan cycle is aborted without updating the beam position. In most cases aborted conscans can be tolerated as long as a minimum of 1 out of ~10 scans succeeds. Thermal drifts within the lidar system are generally slow, and even a reduced update rate is sufficient to track the laser beam. Fast beam tracking is only needed during the beginning of the lidar operation when the

initial position of the laser beam may be far from the optimal position. But even for those cases we find that our conscan implementation achieves nearly perfect beam overlap within 3-6 min. In challenging conditions, e.g. broken clouds, that period may extend up to $\sim 10$ min.

The performance of our conscan implementation may be assessed from Fig. 1 panels a and b. Within the first hour of lidar operation, the FOV of the telescope drifts by more than 150 μrad which is about half of the FOV. Without beam tracking, we would expect a significant decrease of the lidar return signal (shown in panel b) over time. However, peaks in panel b between 4 and 5 UTC indicate a near constant maximum signal of approximately $8 \times 10^4$, a strong indication that the conscan algorithm kept the laser beam centered within the telescope FOV. These peak signals were acquired when the laser beam passed through holes in the cloud layer, while in between the lidar return signal was strongly fluctuating due to scattering within clouds.

We decided to use a sinusoidal modulation of mirror angles because the resulting motion does not require strong accelerations and therefore results in minimal stress on the mirror. Gawronski and Craparo (2002) studied rosette and Lissajous figures in addition to the conscan within the context of spacecraft tracking. In lidar applications, these more complex pattern may also provide advantages in some cases, but a detailed study is needed to assess their potential negative impacts. Gawronski and Craparo (2002) conclude that all three scan techniques have similar properties in the estimation of signal positions and they suggest to use the conscan because of its simplicity.

Actually, our CORAL system uses a bistatic transmitter/receiver configuration. We have never tested the conscan method in a coaxial setup, but there is no obvious reason why it shouldn't work. The algorithm does not assume any particular geometry and the sole requirement is that full geometric overlap between laser and telescope FOV is possible at the altitude where the conscan signal is evaluated.
**8   Page 15 section example. In this section comparisons with other observations are required. No additional observations will fully validate the full profile except radiosondes. Also, many comparisons between other Rayleigh lidars and satellite instruments have been performed, is CORAL found similar deviations. Authors mentioned MLS and SABER, recent comparisons were published and can used as comparisons. The main point will be about the demonstration that such alignment does not introduce any bias. Comparison with radiosonde and temperature retrieval right after alignment will be a first demonstration. Within NDACC, lidars ensure their qualification by comparison with a mobile system running by NASA. Many publications have reported about these comparisons. It is out of the scope of this studies but collocated measurements with other lidars will allow to convinced the scientific community about the data quality.**

We agree that validation of measurement data is an important step. Unfortunately, a second independent mobile lidar system is not available for cross-validation at the location of CORAL. We show a comparison with SABER and ECMWF profiles in Fig. 8 of our manuscript.

A coincident radiosonde sounding, shown in Fig. 2, was acquired during the Southern hemisphere Transport, Dynamics and Chemistry (SOUTHTRAC) campaign. A $\sim$35 min hole in the cloud layer allowed the retrieval of a temperature profile with 20 min integration time centered at 02:10 UTC on 16 November 2019. The period coincides with the upper part of a radiosonde launched at Rio Grande airport approximately 2 km away from CORAL. Both profiles show a nearly perfect agreement with $< 0.6$ K mean difference between 28 and 33 km altitude. While differences increase to $\sim 2.7$ K at the bottom of the lidar profile, we have to keep in mind that the radiosonde crossed these altitudes about 1 hr earlier and atmospheric conditions may have changed. The lidar measurement was acquired in fully automatic mode, demonstrating that the conscan

method achieved beam overlap and tracking in less than 10 minutes after a cold start of the system.

When writing the manuscript we initially did not include this comparison because, contrary to the statement of the reviewer, radiosondes can validate only the lower part of a lidar temperature profile, as Radiosondes rarely reach altitudes $> 35$ km. We will include Fig. 2 in the revised manuscript.

**References**

Ehard, B., Malardel, S., Dörnbrack, A., Kaifler, B., Kaifler, N., and Wedi, N.: Comparing ECMWF high-resolution analyses with lidar temperature measurements in the middle atmosphere, Quarterly Journal of the Royal Meteorological Society, 144, 633–640, https://doi.org/10.1002/qj.3206, https://rmets.onlinelibrary.wiley.com/doi/abs/10.1002/qj.3206, 2018.

Fritts, D. C., Smith, R. B., Taylor, M. J., Doyle, J. D., Eckermann, S. D., Dörnbrack, A., Rapp, M., Williams, B. P., Pautet, P.-D., Bossert, K., Criddle, N. R., Reynolds, C. A., Reinecke, P. A., Uddstrom, M., Revell, M. J., Turner, R., Kaifler, B., Wagner, J. S., Mixa, T., Kruse, C. G., Nugent, A. D., Watson, C. D., Gisinger, S., Smith, S. M., Lieberman, R. S., Laughman, B., Moore, J. J., Brown, W. O., Haggerty, J. A., Rockwell, A., Stossmeister, G. J., Williams, S. F., Hernandez, G., Murphy, D. J., Klekociuk, A. R., Reid, I. M., and Ma, J.: The Deep Propagating Gravity Wave Experiment (DEEPWAVE): An Airborne and Ground-Based Exploration of Gravity Wave Propagation and Effects from Their Sources throughout the Lower and Middle Atmosphere, Bulletin of the American Meteorological Society, 97, 425–453, https://doi.org/10.1175/BAMS-D-14-00269.1, https://doi.org/10.1175/BAMS-D-14-00269.1, 2016.

Gawronski, W. and Craparo, E. M.: Antenna scanning techniques for estimation of spacecraft position, IEEE Antennas and Propagation Magazine, 44, 38–45, https://doi.org/10.1109/MAP.2002.1167263, 2002.

Innis, J. L., Cunningham, A. P., Graham, A. D., and Klekociuk, A. R.: Automatically guiding a telescope to a laser beam on a biaxial antarctic light detection and ranging system, Optical Engineering, 46, 1 – 8, https://doi.org/10.1117/1.2801411, https://doi.org/10.1117/1.2801411, 2007.

Kaifler, B., Lübken, F.-J., Höffner, J., Morris, R. J., and Viehl, T. P.: Lidar observations of gravity wave activity in the middle atmosphere over Davis (69°S, 78°E), Antarctica, Journal of Geophysical Research: Atmospheres, 120, 4506–4521, https://doi.org/10.1002/2014JD022879, https://agupubs.onlinelibrary.wiley.com/doi/abs/10.1002/2014JD022879, 2015.

Li, T., Leblanc, T., McDermid, I. S., Wu, D. L., Dou, X., and Wang, S.: Seasonal and interannual variability of gravity wave activity revealed by long-term lidar observations over Mauna Loa Observatory, Hawaii, Journal of Geophysical Research: Atmospheres, 115, https://doi.org/https://doi.org/10.1029/2009JD013586, https://agupubs.onlinelibrary.wiley.com/doi/abs/10.1029/2009JD013586, 2010.

Mzé, N., Hauchecorne, A., Keckhut, P., and Thétis, M.: Vertical distribution of gravity wave potential energy from long-term Rayleigh lidar data at a northern middle-latitude site, Journal of Geophysical Research: Atmospheres, 119, 12,069–12,083, https://doi.org/https://doi.org/10.1002/2014JD022035, https://agupubs.onlinelibrary.wiley.com/doi/abs/10.1002/2014JD022035, 2014.

Rauthe, M., Gerding, M., and Lübken, F.-J.: Seasonal changes in gravity wave activity measured by lidars at mid-latitudes, Atmospheric Chemistry and Physics, 8, 6775–6787, https://doi.org/10.5194/acp-8-6775-2008, https://acp.copernicus.org/articles/8/6775/2008/, 2008.

Sivakumar, V., Rao, P. B., and Bencherif, H.: Lidar observations of middle atmospheric gravity wave activity over a low-latitude site (Gadanki, 13.5° N, 79.2° E), Annales Geophysicae, 24, 823–834, https://doi.org/10.5194/angeo-24-823-2006, https://angeo.copernicus.org/articles/24/823/2006/, 2006.

Wilson, R., Chanin, M. L., and Hauchecorne, A.: Gravity waves in the middle atmosphere observed by Rayleigh lidar: 2. Climatology, Journal of Geophysical Research: Atmospheres, 96, 5169–5183, https://doi.org/https://doi.org/10.1029/90JD02610, https://agupubs.onlinelibrary.wiley.com/doi/abs/10.1029/90JD02610, 1991.

[Figure]

[Figure]

**Fig. 1.** Performance of the conscan system during the measurement on 3 November 2019, 4-8 UTC. (a) Scan mirror angles, (b) lidar return signal, (c) phase and (d) amplitude of the demodulated signal.

**Fig. 2.** Radiosonde and lidar profile acquired on 16 November 2019. The arrows indicate the height of the radiosonde at different times. All times are mm:hh:ss UTC.

[Figure]

---

## Author Response (AR1)

**Author's Response**

**'A Compact Rayleigh Autonomous Lidar (CORAL) for the middle atmosphere' by Kaifler and Kaifler**

We thank the reviewers for their comments and suggestions which helped us to improve the manuscript. The comments of the two reviewers are reproduced in blue color below with author's responses in black. The revised manuscript with changes highlighted is attached at the end of this document.

**Reviewer #1**

**General comments**

This paper describes a novel autonomous lidar for temperature retrievals in the middle atmosphere that will assist studies of atmospheric processes on a wide variety of timescales. The description of the architecture and methods employed conveys that the instrument is an advance for the field and is able to enhance opportunities for data collection through automation. This description will be of interest to other workers in the field of middle atmosphere lidar by providing a practical basis on which to tackle the vexed issue of maximising the scientific return from high power-aperture product lidars.

I recommend the work for publication after the authors address the specific and technical comments below. These comments are made from the perspective of someone familiar with the practicalities of middle atmosphere lidar in order to engender into the manuscript additional information to more adequately describe the system and methods employed. The standard of presentation and grammar of the manuscript is very good, the figures and tables are appropriate and concise, and the methods used and their description are sound. Overall, I regard that addressing the following comments will constitute a minor revision of the manuscript as significant rewriting and figure changes are not needed.

**Specific comments**

Abstract: The third sentence indicates that the first studies with CORAL show the impact of a strong gravity wave event on stratospheric circulation. I initially took this to mean that this paper included this analysis, but after re-reading the Discussion section I see that this is done in earlier papers. I suggest that the 3rd sentence of the abstract be modified to make this clearer (e.g 'First studies using CORAL data have shown for example...).

Sentence changed to 'First studies using CORAL data have shown for example...'

L87. Please quantify the typical shot-to-shot beam pointing stability of the laser beam after divergence and comment on how this compares with the field of view of the telescope and the divergence of the transmitted beam.

We added '...which reduces the beam divergence to approximately 170 µrad and the pointing jitter to <50 µrad full angle. The resulting effective beam divergence is thus <220 µrad or approximately half of the telescope field of view (FOV).'

L102. What is the blocking level of your Raman filter at 532 nm?

Author's Response: A Compact Rayleigh Autonomous Lidar (CORAL) for the middle atmosphere

We added 'out of band blocking optical depth (OD) >6' and 'The dichroic mirror has a transmission of 1.2% at 532 nm wavelength and this results in a total blocking of the elastic scattering in the Raman channel of OD  $\sim$ 8.'

Section 2.2. Given that the arrangement of the receiver and transmitter are bistatic and that the field of view of the telescope is small (370  $\mu$ rad), the effect of any significant change in focus will depend on how the response function of the field of view changes and the response function of the field of view. How is the field of view of the telescope practically determined (theoretically or measured)? Please comment on the stability of the telescope focus (e.g. the significance of changes due to thermal effects) and how you determine best focus. Please also comment on if and how you can determine the significance of any range-dependent effect on the retrieved temperature profiles due to focus change. L98 (or thereabouts). Please indicate what the expected altitude for full overlap between the fields of view of the transmitter and receiver (presumably below 14 km).

We made a detailed analysis of a thermally-induced focus shift and its impact on fiber coupling. This analysis is documented in the supporting information. We added following sentences in the manuscript:

'The use of carbon fiber tubes results in a high thermal stability of the telescope. A 50 K change in temperature causes the focal point to shift by 160 µm in the vertical. As shown in the supporting information, this shift has as a negligible impact on the fiber coupling performance and retrieved temperature profiles. During setup of the instrument, the optimum position is determined by slowly moving the fiber end up and down using the motor and recoding the strength of the lidar return signal integrated over the altitude range 40-50 km as function of motor position. Repeated scans are performed and measurements are averaged in order to reduce the effect of potential changes in atmospheric transmission during the scans. The focal position is determined as the position with the maximum lidar signal. '

*With the bistatic setup full geometric overlap between the laser beam and the telescope* FOV is achieved at approximately 5 km altitude. However, range-dependent defocusing of the telescope causes the overlap function to vary slightly with altitude. This variable overlap results in a bias in retrieved temperature profiles of less than -0.4 K for altitudes above 40 km and increases to -0.95 K at 15 km altitude (see Section 2 in the supporting information).'

Table 1. What is the quality of the surface of the telescope in terms of RMS surface deviation (in wavelengths) from an ideal paraboloid?

We do not have any measurements of the RMS surface deviation, but we added the information about the spot size in Table 1.

Section 2.3.1. Given that the arrangement of the receiver and transmitter are bistatic, I expect that clouds cause the conscan method to optimise the overlap for lower altitudes. Is that correct and might this cause any issues with data collection? Do you use any data in cases of cloud (even thin cloud)?

Actually, the conscan method appears to be fairly robust with regard to clouds. One major benefit over the more widely used imaging method described in Innis et al. (2007) is that the lidar return signal is analyzed well above potential cloud layers. In our implementation we evaluate the signal between 45 km and 55 km altitude. Though clouds passing through the lidar beam modulate the strength of the lidar return signal and thus impact the demodulation of the conscan error signal, this does not change the altitude for which the beam overlap is optimized. For that reason, we trust our lidar measurements and use the data in case of thin clouds. Thick clouds cause the lidar return signal to drop below a threshold and raw data

profiles (we use 10 s intervals for quality checks) are automatically discarded in the temperature retrieval.

We added three new panels to Fig. 7 in the manuscript. These panels show the phase, amplitude and  $\chi$ 2-value determined from the fits.

*Fig. 7: Performance of the conscan system during the measurement on 3 November 2019.* (*a*) *Mean scan mirror angles, (b) lidar return signal integrated between 45 km and 55 km altitude, (c) phase and (d) amplitude of the demodulated conscan signal, and (e) goodness of the fit.*

We added the following text at the end of Section 2.3.1:

Panels c-e of Fig. 7 show the phase, amplitude and  $\chi^2$ -value determined from the fits to the demodulated conscan signal. The  $\chi$ 2-value is used as indicator for the quality of the conscan signal. Note that we do not normalize the conscan signal prior to fitting the data and hence, the average  $\chi^2$ -value is about 20 instead of unity even for signals with a large signal-to-noise ratio. A large x2-value, we use 300 as a threshold, indicates that the conscan signal could not be properly demodulated and the retrieved estimates of phase and amplitude are not reliable. In this case the conscan cycle is aborted and the beam pointing not updated. In cloudy conditions as many as 9 out of 10 conscans may fail in that way, but the succeeding conscans are still sufficient for beam tracking, as thermal drifts happen on relatively large time scales. If more than 10 successive conscans fail, subsequent intervals are marked in the raw data files of the lidar as potentially having incomplete beam overlap and are discarded in the temperature retrieval. Moreover, beam pointing increments and conscan parameters are stored in raw data files for offline analysis. As evident from Fig. 7e only few conscan measurements exceed the threshold of  $\chi^2 > 300$  even though the lidar return signal was strongly impacted by clouds in the period 4-5 UTC. The larger uncertainties in that period are also reflected in the larger amplitude estimates (Fig. 7d).'

'It is important to note that conscan always evaluates the lidar signal within a constant altitude range in the stratosphere. This is in particular beneficial for bistatic lidars where in an autoguiding setup (e.g. Innis et al. 2007) the camera images the laser beam at an angle in the lower troposphere, as scattering in the stratosphere is too weak for imaging. Due to the finite exposure time, the camera integrates the beam profile along a certain height interval. If a cloud drifts through the beam within this height range, the image will be dominated by the strong Mie scattering. If that happens at the bottom (top) of the height range, camera based autoguiding optimizes the beam overlap for low (high) altitudes in bistatic lidar configurations. The use of conscan avoids this problem.

Section 2.3.1. While you have a cloud-monitoring camera (discussed later) it would seem that introduction of a pellicle beamsplitter in the telescope before the fiber would allow you to monitor the quality and stability of the received beam (e.g. as done in Innis et al., 2007 - https://doi.org/10.1117/1.2801411). Could you please comment on the usefulness or otherwise of such an arrangement for data quality control.

We choose not to use a beam tracking camera as it was done in Innis et al. (2007) for two main reasons. First, pellicle beam splitters are extremely fragile and are unlikely to survive for long periods in a humid environment. CORAL is located close to the coast and we operate it in marginal weather conditions such as thin fog. A plate beamsplitter may solve this problem, but complicates the optical setup. Second, the addition of a beam splitter increases signal losses i.e. less light is available to the receiver. With CORAL we wanted to demonstrate that even small lidars (physical size and laser power) can produce good measurements in the mesosphere if they are optimized for low signal losses.

For some time we used a beam tracking camera looking through a separate Galilei telescope (100 mm aperture) bolted to the main telescope. We found that the error signal determined from the camera was less reliable in (partially) cloudy conditions as compared to the conscan method. We never used the camera signal to assess the data quality because we are confident in the performance of the conscan method (see our comments in the previous section).

Figure 8b. What is the range and time separation for the SABER profile relative to the lidar measurement? Why not show a measurement centred on the time of the ECMWF profile, or is the SABER profile more coincident? What is the source of the ECMWF profile (forecast or analysis) and what is the horizontal grid resolution of the ECMWF data and separation from the measurement site?

The closest SABER profile was acquired at 525 km mean distance to the west of CORAL at 05:38 UTC. We initially chose to start the lidar integration period at the beginning of the hour, but agree that centering it at the time of the SABER profile is more appropriate. Fig. 4 shows the temperature profiles for the updated time period. We use operational ECMWF forecast data with a grid resolution of 0.125°. The grid point closest to the location of CORAL (53.785°S, 67.752°W) is 53.75°S, 67.75°W with a distance of 3.9 km.

We updated the figure with lidar profiles centered at 05:35 UTC and added following text to the caption of Fig. 8:

...SABER profile (acquired at 525 km distance mean distance to the west of CORAL at 05:38 UTC) and 06:00 UTC ECMWF IFS profile (0.125° resolution, grid point 53.75°S 67.75°W, 3.9 km distance to CORAL)...

Section 4. Given that you can be seeding at altitudes of 100 km or above (e.g. Fig. 12b and line355), do you take into account the change in the mean molecular mass of the air in the MLT region, and if so, how?

No, we don't take any change in mean molecular mass into account. This is a short-coming of our retrieval and will be addressed in the future.

Figure 12. Is the seeding altitude at the top of the retrievals in panels (b) and (c) or has allowance been made for convergence of the retrieval?

The profiles shown in Fig. 12b,c of the manuscript start 2 km below the seeding altitude. Normally you would integrate at least one density scale height to allow for the retrieved temperature profile to converge. However, as our group mainly focuses on gravity wave studies and thus predominantly looks at perturbations with mean background temperature profiles subtracted, reaching higher altitudes is of higher importance than absolute convergence. Please note that the 2 km refers to the retrieved profile of the current level within the retrieval pyramid. For example, the 20 min profiles shown in Fig.12c. start at ~90 km which is more than one density scale height below the altitude where the nightly mean profile was seeded with the SABER profile. Hence, we can assume that the seeding temperatures we used for seeding of the 20 min temperature profiles 2 km above are already close to the true atmospheric temperature, allowing for rapid convergence.

L352. You are describing your retrieval method in fairly general terms and I would recommend that you are more specific, referencing earlier work as necessary or expanding Section 4 to more fully explain the particular criteria and assumptions that you are using. What are the coincidence criteria that you apply in the use of satellite data for seeding? Do you account for bias in the seed temperature? The averaging kernels of the MLS measurements are coarse compared to SABER in the MLT. When would you use MLS measurements to seed the profiles? How do you determine what effective height to assign to the seeding temperature obtained from MLS given that the vertical resolution of the lidar data is much less than the averaging kernel of the MLS retrieval (8-10 km)? Is the 100-108 km altitude range within the recommended upper limit for scientifically useful MLS data at your site. Is there a suitable MLS profile you can show in Fig. 8?

We assumed that the hydrostatic temperature retrieval is sufficiently described in literature and discussed only the non-standard parts specific to our implementation. However, we agree that this section should be expanded to benefit a larger audience. We added the following text at the start of section 4:

'In the absence of aerosols and absorption by trace gases such as ozone, and after taking into account the geometric factor  $z^2$  stemming from the formation of spherical waves of scattered light, the lidar return signal can be assumed as proportional to the Rayleigh backscatter of the atmosphere (e.g. Leblanc et al., 1998). Because Rayleigh scattering is directly proportional to the number density of air, the Rayleigh backscatter profile equates to a profile of relative number density. In the retrieval the atmosphere is split up in discrete layers and the relative number density is evaluated at these layers from layer-averages of the Rayleigh backscatter profile assuming that the number density varies exponentially with altitude across layers. Using the ideal gas law and assuming hydrostatic equilibrium, temperatures are calculated by integrating relative number densities from top to bottom. One complication arises from the requirement to start the integration at infinity where the density is zero. Since this is impractical because of the large noise in lidar measurements at very high altitudes, instead, the integration is split into two parts. Above a certain reference altitude  $z_0$  the integration is assumed to evaluate to a reference temperature  $T_0$  that is then used to initialize or 'seed' the integration of the second part starting at  $z_0$ . Typically,  $z_0$  is chosen as the upper boundary of the lidar return profile where the signal is still reliable, and the corresponding  $T_0$  is taken from other independent measurements or models (e.g. Duck et al., 2001; Alexander et al., 2011). The condition 'no aerosols present' can be relaxed if inelastic (Raman) scattering e.g. at nitrogen molecules is used instead of the elastic scattering to derive profiles of relative number density."

As coincidence criteria we chose a maximum horizontal distance of 1000 km between the mean location of the satellite measurement and the location of CORAL and a maximum separation in time of 12 h. Normally, we used SABER temperature profiles for seeding, but for rare cases when SABER data are not available, e.g. when there were problems with the instrument, we did fall back to MLS profiles (accounted for ~4% of all seed profiles). However, during revision of the manuscript we followed up on the issue raised by the

reviewer and revisited literature on MLS observations. This led us to conclude that MLS profiles are not really suitable for seeding the temperature integration in the upper MLT region. Following that we removed the MLS profiles from the seeding data and now fall back to the closest SABER profile in time, either before or after the lidar observation, if no coincident SABER profile is available. We updated the retrieval section in order to reflect that change.

**We added the sentence:**

'As coincidence criteria we chose a maximum horizontal distance between the mean location of the satellite measurement and the location of CORAL.'

L360. Can you please indicate what the SNR threshold for the far-channel corresponds to as a relative uncertainty in pseudo-density (e.g. section 1.2 in Wing et al., 2020 - https://doi.org/10.3390/atmos11010075 and Alexander et al., 2011 - https://doi.org/10.1029/2010JD015164 who use 20% and 10% for their seeding threshold, respectively)?

Our criterion of SNR>4 in relation to 100 m bin width translates to a relative uncertainty in pseudo-density of 8.3 % for the default vertical resolution of 900 m and 6.5 % for 1500 m resolution.

**We added the following sentences:**

'The criterion SNR > 4 at 100 m bin width translates to a relative uncertainty in pseudodensity of 8.3 % in case of the default resolution of 900 m and 6.5 % for 1500 m resolution. For reference, a threshold of 10 % was used by Alexander et al. (2011).'

**L379. By reference to earlier analysis of CORAL data, please indicate if any significant tidal signatures are present in the MLT data in Fig 12c and what characteristics of the temperature variations allow you to conclude a 5-7 hour period internal gravity wave is present (e.g. downward phase progressions or narrow duty cycle, perhaps highlighting with dashed lines).**

We tried to investigate tides using the CORAL data set but had very limited success. The two main problems are that in winter temperature variability is dominated by mountain waves, and in summer, when gravity wave amplitudes are low, measurements are not long enough to capture tides. Measurements in spring indicate amplitudes of the semidiurnal tide of